# THE IMPLICIT BIAS OF GRADIENT DESCENT ON SEPARABLE DATA

**Daniel Soudry, Elad Hoffer, Mor Shpigel Nacson**
Department of Electrical Engineering,Technion
Haifa, 320003, Israel
daniel.soudry@gmail.com
elad.hoffer@gmail.com
mor.shpigel@gmail.com

**Nathan Srebro**
Toyota Technological Institute at Chicago
Chicago, Illinois 60637, USA
nati@ttic.edu

## ABSTRACT

We show that gradient descent on an unregularized logistic regression problem, for almost all separable datasets, converges to the same direction as the max-margin solution. The result generalizes also to other monotone decreasing loss functions with an infimum at infinity, and we also discuss a multi-class generalizations to the cross entropy loss. Furthermore, we show this convergence is very slow, and only logarithmic in the convergence of the loss itself. This can help explain the benefit of continuing to optimize the logistic or cross-entropy loss even after the training error is zero and the training loss is extremely small, and, as we show, even if the validation loss increases. Our methodology can also aid in understanding implicit regularization in more complex models and with other optimization methods.

## 1 INTRODUCTION

It is becoming increasingly clear that implicit biases introduced by the optimization algorithm play a crucial role in deep learning and in the generalization ability of the learned models (Neyshabur et al., 2014; 2015; Zhang et al., 2017; Keskar et al., 2017; Neyshabur et al., 2017; Wilson et al., 2017). In particular, minimizing the training error, without any explicit regularization, over models with more parameters and more capacity then the number of training examples, often yields good generalization, despite the empirical optimization problem being highly underdetermined. That is, there are many global minima of the training objective, most of which will not generalize well, but the optimization algorithm (*e.g.* gradient descent) biases us toward a particular minimum that *does* generalize well. Unfortunately, we still do not have a good understanding of the biases introduced by different optimization algorithms in different situations.

We do have a decent understanding of the implicit regularization introduced by early stopping of stochastic methods or, at an extreme, of one-pass (no repetition) stochastic optimization. However, as discussed above, in deep learning we often benefit from implicit bias even when optimizing the (unregularized) training error to convergence, using stochastic or batch methods. For loss functions with attainable, finite, minimizers, such as the squared loss, we have some understanding of this: In particular, when minimizing an underdetermined least squares problem using gradient descent starting from the origin, we know we will converge to the minimum Euclidean norm solution. But the logistic loss, and its generalization the cross-entropy loss which is often used in deep learning, do not admit a finite minimizer on separable problems. Instead, to drive the loss toward zero and thus minimize it, the predictor must diverge toward infinity.

Do we still benefit from implicit regularization when minimizing the logistic loss on separable data? Clearly the norm of the predictor itself is not minimized, since it grows to infinity. However, for prediction, only the direction of the predictor, *i.e.* the normalized $\mathbf{w}(t)/\|\mathbf{w}(t)\|$, is important. How does $\mathbf{w}(t)/\|\mathbf{w}(t)\|$ behave as $t \to \infty$ when we minimize the logistic (or similar) loss using gradient descent on separable data, *i.e.*, when it is possible to get zero misclassification error and thus drive the loss to zero?

In this paper, we show that even without any explicit regularization, for all most all datasets (except a zero measure set), when minimizing linearly separable logistic regression problems using gradient

descent, we have that $\mathbf{w}(t)/\|\mathbf{w}(t)\|$ converges to the $L_2$ maximum margin separator, *i.e.* to the solution of the hard margin SVM. This happens even though the norm $\|\mathbf{w}\|$, nor the margin constraint, are in no way part of the objective nor explicitly introduced into optimization. More generally, we show the same behavior for generalized linear problems with any smooth, monotone strictly decreasing, lower bounded loss with an exponential tail. Furthermore, we characterize the rate of this convergence, and show that it is rather slow, with the distance to the max-margin predictor decreasing only as $O(1/\log(t))$. This explains why the predictor continues to improve even when the training loss is already extremely small. We emphasize and demonstrate that this bias is specific to gradient descent, and changing the optimization algorithm, *e.g.* using adaptive learning rate methods such as ADAM Kingma & Ba (2015), changes this implicit bias.

## 2 MAIN RESULTS

Consider a dataset $\{\mathbf{x}_n, y_n\}_{n=1}^N$, with binary labels $y_n \in \{-1, 1\}$. We analyze learning by minimizing an empirical loss of the form

$$\mathcal{L}(\mathbf{w}) = \sum_{n=1}^N \ell\left(y_n \mathbf{w}^\top \mathbf{x}_n\right) . \tag{2.1}$$

where $\mathbf{w} \in \mathbb{R}^d$ is the weight vector. A bias term could be added in the usual way, extending $\mathbf{x}_n$ by an additional '1' component. To simplify notation, we assume that $\forall n : y_n = 1$ — this is true without loss of generality, since we can always re-define $y_n \mathbf{x}_n$ as $\mathbf{x}_n$.

We are particularly interested in problems that are linearly separable, and the loss is smooth monotone strictly decreasing and non-negative:

**Assumption 1.** *The dataset is strictly linearly separable:* $\exists \mathbf{w}_*$ *such that* $\forall n : \mathbf{w}_*^\top \mathbf{x}_n > 0$ .

**Assumption 2.** $\ell(u)$ *is a positive, differentiable, monotonically decreasing to zero*[1]*, (so* $\forall u : \ell(u) > 0, \ell'(u) < 0$ *and* $\lim_{u\to\infty} \ell(u) = \lim_{u\to\infty} \ell'(u) = 0$) *and a* $\beta$-*smooth function,* i.e. *its derivative is* $\beta$-*Lipshitz.*

Many common loss functions, including the logistic, exp-loss, probit and sigmoidal losses, follow Assumption 2. Assumption 2 also straightforwardly implies that $\mathcal{L}(\mathbf{w})$ is a $\beta\sigma_{\max}^2(\mathbf{X})$-smooth function, where the columns of $\mathbf{X}$ are all samples, and $\sigma_{\max}(\mathbf{X})$ is the maximal singular value of $\mathbf{X}$.

Under these conditions, the infimum of the optimization problem is zero, but it is not attained at any finite $\mathbf{w}$. Furthermore, no finite critical point $\mathbf{w}$ exist. We consider minimizing eq. 2.1 using Gradient Descent (GD) with a fixed learning rate $\eta$, *i.e.,* with steps of the form:

$$\mathbf{w}(t+1) = \mathbf{w}(t) - \eta \nabla \mathcal{L}(\mathbf{w}(t)) = \mathbf{w}(t) - \eta \sum_{n=1}^N \ell'\left(\mathbf{w}(t)^\top \mathbf{x}_n\right) \mathbf{x}_n. \tag{2.2}$$

We do not require convexity. Under Assumptions 1 and 2, gradient descent converges to the global minimum (*i.e.* to zero loss) even without it:

**Lemma 1.** *Let* $\mathbf{w}(t)$ *be the iterates of gradient descent (eq. 2.2) with* $\eta < 2\beta^{-1}\sigma_{\max}^{-2}(\mathbf{X})$ *and any starting point* $\mathbf{w}(0)$. *Under Assumptions 1 and 2, we have: (1)* $\lim_{t\to\infty} \mathcal{L}(\mathbf{w}(t)) = 0$, *(2)* $\lim_{t\to\infty} \|\mathbf{w}(t)\| = \infty$, *and (3)* $\forall n : \lim_{t\to\infty} \mathbf{w}(t)^\top \mathbf{x}_n = \infty$.

*Proof.* Since the data is strictly linearly separable, $\exists \mathbf{w}_*$ which linearly separates the data, and therefore

$$\mathbf{w}_*^\top \nabla \mathcal{L}(\mathbf{w}) = \sum_{n=1}^N \ell'\left(\mathbf{w}^\top \mathbf{x}_n\right) \mathbf{w}_*^\top \mathbf{x}_n.$$

For any finite $\mathbf{w}$, this sum cannot be equal to zero, as a sum of negative terms, since $\forall n : \mathbf{w}_*^\top \mathbf{x}_n > 0$ and $\forall u : \ell'(u) < 0$. Therefore, there are no finite critical points $\mathbf{w}$, for which $\nabla \mathcal{L}(\mathbf{w}) = \mathbf{0}$. But

---

[1]The requirement of nonnegativity and that the loss asymptotes to zero is purely for convenience. It is enough to require the loss is monotone decreasing and bounded from below. Any such loss asymptotes to some constant, and is thus equivalent to one that satisfies this assumption, up to a shift by that constant.

gradient descent on a smooth loss with an appropriate stepsize is always guaranteed to converge to a critical point: $\nabla \mathcal{L}(\mathbf{w}(t)) \to \mathbf{0}$ (see, *e.g.* Lemma 5 in Appendix A.4, slightly adapted from Ganti (2015), Theorem 2). This necessarily implies that $\|\mathbf{w}(t)\| \to \infty$ while $\forall n: \mathbf{w}(t)^\top \mathbf{x}_n > 0$ for large enough $t$—since only then $\ell'\left(\mathbf{w}(t)^\top \mathbf{x}_n\right) \to 0$. Therefore, $\mathcal{L}(\mathbf{w}) \to 0$, so GD converges to the global minimum. $\qquad\square$

The main question we ask is: can we characterize the direction in which $\mathbf{w}(t)$ diverges? That is, does the limit $\lim_{t\to\infty} \mathbf{w}(t) / \|\mathbf{w}(t)\|$ always exists, and if so, what is it?

In order to analyze this limit, we will need to make a further assumption on the tail of the loss function:

**Definition 2.** *A function $f(u)$ has a "tight exponential tail", if there exist positive constants $c, a, \mu_+, \mu_-, u_+$ and $u_-$ such that*

$$\forall u > u_+ : f(u) \le c\left(1 + \exp\left(-\mu_+ u\right)\right) e^{-au}$$
$$\forall u > u_- : f(u) \ge c\left(1 - \exp\left(-\mu_- u\right)\right) e^{-au}.$$

**Assumption 3.** *The negative loss derivative $-\ell'(u)$ has a tight exponential tail (Definition 2).*

For example, the exponential loss $\ell(u) = e^{-u}$ and the commonly used logistic loss $\ell(u) = \log\left(1 + e^{-u}\right)$ both follow this assumption with $a = c = 1$. We will assume $a = c = 1$ — without loss of generality, since these constants can be always absorbed by re-scaling $\mathbf{x}_n$ and $\eta$.

We are now ready to state our main result:

**Theorem 3.** *For almost all datasets (i.e., except for a measure zero) which are strictly linearly separable (Assumption 1) and given a $\beta$-smooth decreasing loss function (Assumption 2) with an exponential tail (Assumption 3), gradient descent (as in eq. 2.2) with stepsize $\eta < 2\beta^{-1}\sigma_{\max}^{-2}(\mathbf{X})$ and any starting point $\mathbf{w}(0)$ will behave as:*

$$\mathbf{w}(t) = \hat{\mathbf{w}} \log t + \boldsymbol{\rho}(t) , \tag{2.3}$$

*where the residual $\boldsymbol{\rho}(t)$ is bounded and so*

$$\lim_{t\to\infty} \frac{\mathbf{w}(t)}{\|\mathbf{w}(t)\|} = \frac{\hat{\mathbf{w}}}{\|\hat{\mathbf{w}}\|}$$

*where $\hat{\mathbf{w}}$ is the $L_2$ max margin vector (the solution to the hard margin SVM):*

$$\hat{\mathbf{w}} = \underset{\mathbf{w}\in\mathbb{R}^d}{\operatorname{argmin}} \|\mathbf{w}\|^2 \ \text{ s.t. } \ \mathbf{w}^\top \mathbf{x}_n \ge 1. \tag{2.4}$$

Since the theorem holds for almost all datasets, in particular, it holds with probability 1 if $\{\mathbf{x}_n\}_{n=1}^N$ are sampled from an absolutely continuous distribution.

**Proof Sketch** We first understand intuitively why an exponential tail of the loss entail asymptotic convergence to the max margin vector: Assume for simplicity that $\ell(u) = e^{-u}$ exactly, and examine the asymptotic regime of gradient descent in which $\forall n: \mathbf{w}(t)^\top \mathbf{x}_n \to \infty$, as is guaranteed by Lemma 1. If $\mathbf{w}(t) / \|\mathbf{w}(t)\|$ converges to some limit $\mathbf{w}_\infty$, then we can write $\mathbf{w}(t) = g(t)\mathbf{w}_\infty + \boldsymbol{\rho}(t)$ such that $g(t) \to \infty$, $\forall n: \mathbf{x}_n^\top \mathbf{w}_\infty > 0$, and $\lim_{t\to\infty} \boldsymbol{\rho}(t)/g(t) = 0$. The gradient can then be written as:

$$-\nabla \mathcal{L}(\mathbf{w}) = \sum_{n=1}^N \exp\left(-\mathbf{w}(t)^\top \mathbf{x}_n\right)\mathbf{x}_n = \sum_{n=1}^N \exp\left(-g(t)\mathbf{w}_\infty^\top \mathbf{x}_n\right)\exp\left(-\boldsymbol{\rho}(t)^\top \mathbf{x}_n\right)\mathbf{x}_n .$$
$$\tag{2.5}$$

As $g(t) \to \infty$ and the exponents become more negative, only those samples with the largest (*i.e.*, least negative) exponents will contribute to the gradient. These are precisely the samples with the smallest margin $\operatorname{argmin}_n \mathbf{w}_\infty^\top \mathbf{x}_n$, aka the "support vectors". The negative gradient (eq. 2.5) would then asymptotically become a non-negative linear combination of support vectors. The limit $\mathbf{w}_\infty$ will then be dominated by these gradients, since any initial conditions become negligible as $\|\mathbf{w}(t)\| \to \infty$

(from Lemma 1). Therefore, $\mathbf{w}_\infty$ will also be non-negative linear combination of support vectors, and so will its scaling $\hat{\mathbf{w}} = \mathbf{w}_\infty / \left( \min_n \mathbf{w}_\infty^\top \mathbf{x}_n \right)$. We therefore have:

$$\hat{\mathbf{w}} = \sum_{n=1}^{N} \alpha_n \mathbf{x}_n \qquad \forall n \ \left( \alpha_n \geq 0 \text{ and } \hat{\mathbf{w}}^\top \mathbf{x}_n = 1 \right) \ \text{ OR } \ \left( \alpha_n = 0 \text{ and } \hat{\mathbf{w}}^\top \mathbf{x}_n > 1 \right) \qquad (2.6)$$

These are precisely the KKT condition for the SVM problem (eq. 2.4) and we can conclude that $\hat{\mathbf{w}}$ is indeed its solution and $\mathbf{w}_\infty$ is thus proportional to it.

To prove Theorem 3 rigorously, we need to show that $\mathbf{w}(t) / \|\mathbf{w}(t)\|$ has a limit, that $g(t) = \log(t)$ and to bound the effect of various residual errors, such as gradients of non-support vectors and the fact that the loss is only approximately exponential. To do so, we substitute eq. 2.3 into the gradient descent dynamics (eq. 2.2), with $\mathbf{w}_\infty = \hat{\mathbf{w}}$ being the max margin vector and $g(t) = \log t$. We then show that the increment in the norm of $\boldsymbol{\rho}(t)$ is bounded by $C_1 t^{-\nu}$ for some $C_1 > 0$ and $\nu > 1$, which is a converging series. This happens because the increment in the max margin term, $\hat{\mathbf{w}}[\log(t+1) - \log(t)] \approx \hat{\mathbf{w}} t^{-1}$, cancels out the dominant $t^{-1}$ term in the gradient $-\nabla \mathcal{L}(\mathbf{w}(t))$ (eq. 2.5 with $g(t) = \log(t)$ and $\mathbf{w}_\infty^\top \mathbf{x}_n = 1$). A complete proof can be found in Appendix A.

**More refined analysis: characterizing the residual** We can furthermore characterize the asymptotic behavior of $\boldsymbol{\rho}(t)$. To do so, we need to refer to the KKT conditions (eq. 2.6) of the SVM problem (eq. 2.4) and the associated support vectors $\mathcal{S} = \mathrm{argmin}_n \hat{\mathbf{w}}^\top \mathbf{x}_n$. The following refinement of Theorem 3 is also proved in Appendix A:

**Theorem 4.** *Under the conditions and notation of Theorem 3, if, in addition the support vectors span the data (i.e.* $\mathrm{rank}(\mathbf{X}_\mathcal{S}) = \mathrm{rank}(\mathbf{X})$ *where the columns of* $\mathbf{X}$ *are all samples and of* $\mathbf{X}_\mathcal{S}$ *are the support vectors), then* $\lim_{t \to \infty} \boldsymbol{\rho}(t) = \tilde{\mathbf{w}}$, *where* $\tilde{\mathbf{w}}$ *is unique, given* $\mathbf{w}(0)$, *and a solution to*

$$\forall n \in \mathcal{S}: \ \eta \exp\left( -\mathbf{x}_n^\top \tilde{\mathbf{w}} \right) = \alpha_n \qquad (2.7)$$

Note these equations are well-defined for almost all datasets, since (see Lemma 8 in Appendix F) then there are at most $d$ support vectors, $\alpha_n$ are unique and $\forall n \in \mathcal{S}: \alpha_n \neq 0$.

## 3 IMPLICATIONS: RATES OF CONVERGENCE

The solution in eq. 2.3 implies that $\mathbf{w}(t) / \|\mathbf{w}(t)\|$ converges to the normalized max margin vector $\hat{\mathbf{w}} / \|\hat{\mathbf{w}}\|$. Moreover, this convergence is very slow— logarithmic in the number of iterations. Specifically, in Appendix B we show that Theorem 3 implies the following tight rates of convergence:

The normalized weight vector converges to normalized max margin vector in $L_2$ norm

$$\left\| \frac{\mathbf{w}(t)}{\|\mathbf{w}(t)\|} - \frac{\hat{\mathbf{w}}}{\|\hat{\mathbf{w}}\|} \right\| = O(1/\log t) , \qquad (3.1)$$

and in angle

$$1 - \frac{\mathbf{w}(t)^\top \hat{\mathbf{w}}}{\|\mathbf{w}(t)\| \|\hat{\mathbf{w}}\|} = O\left(1/\log^2 t\right) , \qquad (3.2)$$

and the margin converges as

$$\frac{1}{\|\hat{\mathbf{w}}\|} - \frac{\min_n \mathbf{x}_n^\top \mathbf{w}(t)}{\|\mathbf{w}(t)\|} = O(1/\log t) ; \qquad (3.3)$$

this slow convergence is in sharp contrast to the convergence of the (training) loss:

$$\mathcal{L}(\mathbf{w}(t)) = O\left(t^{-1}\right) . \qquad (3.4)$$

A simple construction (also in Appendix B) shows that the rates in the above equations are tight. Thus, the convergence of $\mathbf{w}(t)$ to the max-margin $\hat{\mathbf{w}}$ can be logarithmic in the loss itself, and we might need to wait until the loss is exponentially small in order to be close to the max-margin solution. This can help explain why continuing to optimize the training loss, even after the training error is zero and the training loss is extremely small, still improves generalization performance—our results suggests that the margin could still be improving significantly in this regime.

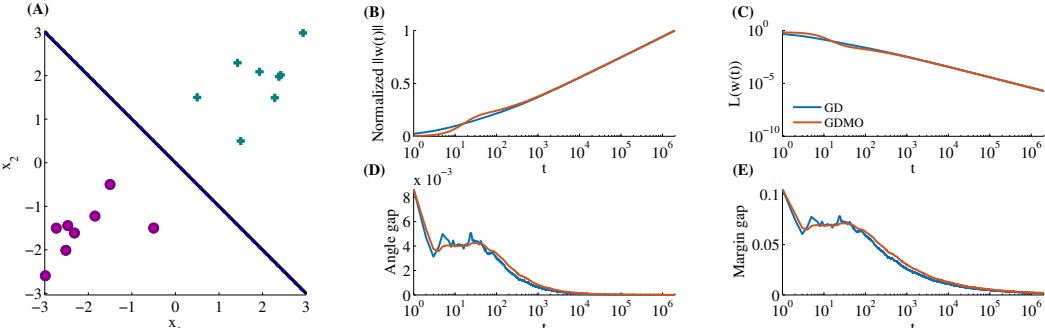

Figure 1: Visualization of or main results on a synthetic dataset in which the $L_2$ max margin vector $\hat{\mathbf{w}}$ is precisely known. **(A)** The dataset (positive and negatives samples ($y = \pm 1$) are respectively denoted by $'+'$ and $'\circ'$), max margin separating hyperplane (black line), and the asymptotic solution of GD (dashed blue). For both GD and GD with momentum (GDMO), we show: **(B)** The norm of $\mathbf{w}(t)$, normalized so it would equal to 1 at the last iteration, to facilitate comparison. As expected (eq. 2.3), the norm increases logarithmically; **(C)** the training loss. As expected, it decreases as $t^{-1}$ (eq. 3.4); and **(D&E)** the angle and margin gap of $\mathbf{w}(t)$ from $\hat{\mathbf{w}}$ (eqs. 3.2 and 3.3). As expected, these are logarithmically decreasing to zero. **Implementation details:** The dataset includes four support vectors: $\mathbf{x}_1 = (0.5, 1.5)$, $\mathbf{x}_2 = (1.5, 0.5)$ with $y_1 = y_2 = 1$, and $\mathbf{x}_3 = -\mathbf{x}_1$, $\mathbf{x}_4 = -\mathbf{x}_2$ with $y_3 = y_4 = -1$ (the $L_2$ normalized max margin vector is then $\hat{\mathbf{w}} = (1,1)/\sqrt{2}$ with margin equal to $\sqrt{2}$), and 12 other random datapoints (6 from each class), that are not on the margin. We used a learning rate $\eta = 1/\sigma_{\max}(\mathbf{X})$, where $\sigma_{\max}(\mathbf{X})$ is the maximal singular value of $\mathbf{X}$, momentum $\gamma = 0.9$ for GDMO, and initialized at the origin.

A numerical illustration of the convergence is depicted in Figure 1. As predicted by the theory, the norm $\|\mathbf{w}(t)\|$ grows logarithmically (note the semi-log scaling), and $\mathbf{w}(t)$ converges to the max-margin separator, but only logarithmically, while the loss itself decreases very rapidly (note the log-log scaling).

An important practical consequence of our theory, is that although the margin of $\mathbf{w}(t)$ keeps improving, and so we can expect the population (or test) misclassification error of $\mathbf{w}(t)$ to improve for many datasets, the same cannot be said about the expected population loss (or test loss)! At the limit, the direction of $\mathbf{w}(t)$ will converge toward the max margin predictor $\hat{\mathbf{w}}$. Although $\hat{\mathbf{w}}$ has zero training error, it will not generally have zero misclassification error on the population, or on a test or a validation set. Since the norm of $\mathbf{w}(t)$ will increase, if we use the logistic loss or any other convex loss, the loss incurred on those misclassified points will also increase. More formally, consider the logistic loss $\ell(u) = \log(1 + e^{-u})$ and define also the hinge-at-zero loss $h(u) = \max(0, -u)$. Since $\hat{\mathbf{w}}$ classifies all training points correctly, we have that on the training set $\sum_{n=1}^N h(\hat{\mathbf{w}}^\top \mathbf{x}_n) = 0$. However, on the population we would expect some errors and so $\mathbb{E}[h(\hat{\mathbf{w}}^\top \mathbf{x})] > 0$. Since $\mathbf{w}(t) \approx \hat{\mathbf{w}} \log t$ and $\ell(\alpha u) \to \alpha h(u)$ as $\alpha \to \infty$, we have:

$$\mathbb{E}[\ell(\mathbf{w}(t)^\top \mathbf{x})] \approx \mathbb{E}[\ell((\log t)\hat{\mathbf{w}}^\top \mathbf{x})] \approx (\log t)\mathbb{E}[h(\hat{\mathbf{w}}^\top \mathbf{x})] = \Omega(\log t). \qquad (3.5)$$

That is, the population loss increases logarithmically while the margin and the population misclassification error improve. Roughly speaking, the improvement in misclassification does not out-weight the increase in the loss of those points still misclassified.

The increase in the test loss is practically important because the loss on a validation set is frequently used to monitor progress and decide on stopping. Similar to the population loss, the validation loss $\mathcal{L}_{\text{val}}(\mathbf{w}(t)) = \sum_{\mathbf{x} \in \mathcal{V}} \ell\left(\mathbf{w}(t)^\top \mathbf{x}\right)$ calculated on an independent validation set $\mathcal{V}$, will increase logarithmically with $t$ (since we would not expect zero validation error), which might cause us to think we are over-fitting or otherwise encourage us to stop the optimization. But this increase does not actually represent the model getting worse, merely $\|\mathbf{w}(t)\|$ getting larger, and in fact the model might be getting better (with larger margin and possibly smaller error rate).

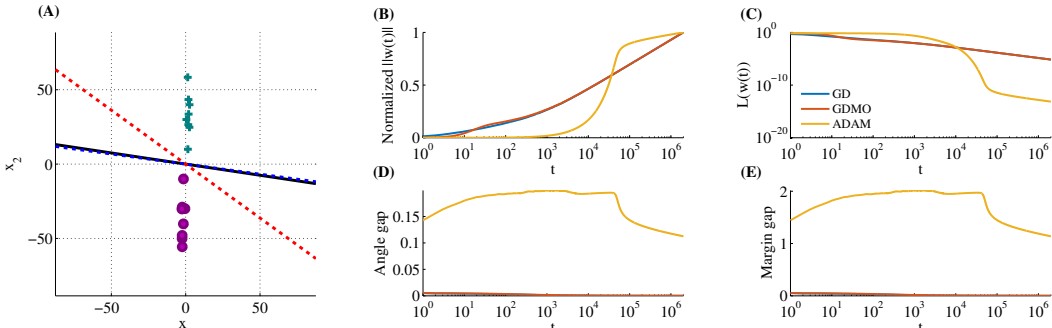

Figure 2: Same as Fig. 1, except we multiplied all $x_2$ values in the dastaset by 20, and also train using ADAM. The final weight vector produced after $2 \cdot 10^6$ epochs of optimization using ADAM (red dashed line) does not converge to L2 max margin solution (black line), in contrast to GD (blue dashed line), or GDMO.

## 4 EXTENSIONS

We discuss several possible extensions of our results.

### 4.1 MULTI-CLASS CLASSIFICATION WITH CROSS-ENTROPY LOSS

So far, we have discussed the problem of binary classification. For multi-class problems commonly encountered, we frequently learn a predictor $\mathbf{w}_k$ for each class, and use the cross-entropy loss with a softmax output, which is a generalization of the logistic loss. What do the linear predictors $\mathbf{w}_k(t)$ converge to if we minimize the cross-entropy loss by gradient descent on the predictors? In Appendix C we analyze this problem for separable data, and show that again, the predictors diverge to infinity and the loss converges to zero. Furthermore, we show that, generically, the loss converges to a logistic loss for transformed data, for which our theorems hold. This strongly suggests that gradient descent converges to a scaling of the $K$-class SVM solution:

$$\arg \min_{\mathbf{w}_1,\ldots,\mathbf{w}_K} \sum_{k=1}^{K} \|\mathbf{w}_k\|^2 \text{ s.t. } \forall n, \forall k \neq y_n : \mathbf{w}_{y_n}^\top \mathbf{x}_n \geq \mathbf{w}_k^\top \mathbf{x}_n + 1 \tag{4.1}$$

We believe this can also be established rigorously and for generic exponential tailed multi-class loss.

### 4.2 OTHER OPTIMIZATION METHODS

In this paper we examined the implicit bias of gradient descent. Different optimization algorithms exhibit different biases, and understanding these biases and how they differ is crucial to understanding and constructing learning methods attuned to the inductive biases we expect. Can we characterize the implicit bias and convergence rate in other optimization methods?

In Figure 1 we see that adding momentum does not qualitatively affects the bias induced by gradient descent. In Figure 4 in Appendix E we also repeat the experiment using stochastic gradient descent, and observe a similar bias. This is consistent with the fact that momentum, acceleration and stochasticity do not change the bias when using gradient descent to optimize an under determined least squares problems. It would be beneficial, though, to rigorously understand how much we can generalize our result to gradient descent variants, and how the convergence rates might change in these cases.

Employing adaptive methods, such as AdaGrad (Duchi et al., 2011) and ADAM (Kingma & Ba, 2015), does significantly affect the bias. In Figure 2 we show the predictors obtained by ADAM and by gradient descent on a simple data set. Both methods converge to zero training error solutions. But although gradient descent converges to the $L_2$ max margin predictor, as predicted by our theory, ADAM does not. The implicit bias of adaptive method has been a recent topic of interest, with Hoffer et al. (2017) and Wilson et al. (2017) suggesting they lead to worse generalization.

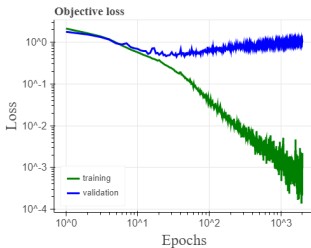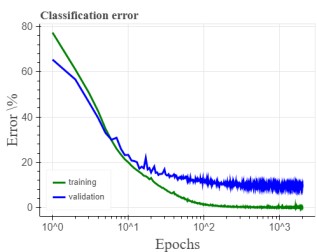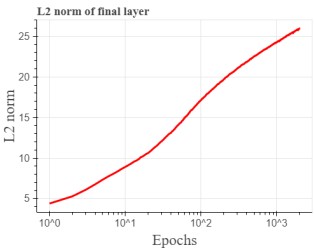

Figure 3: Training of a convolutional neural network on CIFAR10 using stochastic gradient descent with constant learning rate and momentum, softmax output and a cross entropy loss, where we achieve $8.3\%$ final validation error. We observe that, approximately: (1) The training loss decays as a $t^{-1}$, (2) the $L_2$ norm of last weight layer increases logarithmically, (3) after a while, the validation loss starts to increase, and (4) in contrast, the validation (classification) error slowly improves.

Wilson et al. discuss the limit of AdaGrad on lest square problems, but fall short of providing an actual characterization of the limit. This is not surprising, as the limit of AdaGrad on least square problems is fragile and depends on the choice of stepsize and other parameters, and thus complicated to characterize. We expect our methodology could be used to precisely characterize the implicit bias of such methods on logistic regression problems. The asymptotic nature of the analysis is appealing here, as it is insensitive to the initial point, initial conditioning matrix, and large initial steps.

More broadly, it would be interesting to study the behavior of mirror descent and natural gradient descent, and relate the bias they induce to the potential function or divergence underlying them. A reasonable conjecture, which we have not yet investigated, is that for any potential function $\Psi(\mathbf{w})$, these methods converge to the maximum $\Psi$-margin solution $\arg\min_{\mathbf{w}} \Psi(\mathbf{w})\text{s.t.}\forall n : \mathbf{w}^\top \mathbf{x}_n \geq 1$. Since mirror descent can be viewed as regularizing progress using $\Psi(\mathbf{w})$, it is worth noting the results of Rosset et al. (2004b): they considered the regularization path $\mathbf{w}_\lambda = \arg\min \mathcal{L}(\mathbf{w}) + \lambda \|\mathbf{w}\|_p^p$ for similar loss function as we do, and showed that $\lim_{\lambda \to 0} \mathbf{w}_\lambda / \|\mathbf{w}_\lambda\|_p$ is proportional to the maximum $L_p$ margin solution. Rosset et al. do not consider the effect of the optimization algorithm, and instead add explicit regularization—here we are specifically interested in the bias implied by the algorithm *not* by adding (even infinitesimal) explicit regularization.

Our analysis also covers the exp-loss used in boosting, as its tail is similar to that of the logistic loss. However, boosting is a coordinate descent procedure, and not a gradient descent procedure. Indeed, the coordinate descent interpretation of AdaBoost shows that coordinate descent on the exp-loss for a linearly separable problem is related to finding the maximum $L_1$ margin solution (Schapire et al., 1998; Rosset et al., 2004a; Shalev-Shwartz & Singer, 2010).

## 4.3 DEEP NETWORKS

In this paper, we only consider linear prediction. Naturally, it is desirable to generalize our results also to non-linear models and especially multi-layer neural networks.

Even without a formal extension and description of the precise bias, our results already shed light on how minimizing the cross-entropy loss with gradient descent can have a margin maximizing effect, how the margin might improve only logarithmically slow, and why it might continue improving even as the validation loss increases. These effects are demonstrated in Figure 3 and Table 1 which portray typical training of a convolutional neural network using unregularized gradient descent[2]. As can be seen, the norm of the weight increases, but the validation error continues decreasing, albeit very slowly (as predicted by the theory), even after the training error is zero and the training loss is extremely small. We can now understand how even though the loss is already extremely small, some sort of margin might be gradually improving as we continue optimizing. We can also observe how the validation loss increases despite the validation error decreasing, as discussed in Section 3.

As an initial advance toward tackling deep network, we can point out that for two special cases, our results may be directly applied to multi-layered networks. First, our results may be applied exactly,

---

[2]Code available here: `https://github.com/paper-submissions/MaxMargin`

| Epoch | 50 | 100 | 200 | 400 | 2000 | 4000 |
|---|---|---|---|---|---|---|
| $L_2$ norm | 13.6 | 16.5 | 19.6 | 20.3 | 25.9 | 27.54 |
| Train loss | 0.1 | 0.03 | 0.02 | 0.002 | $10^{-4}$ | $3 \cdot 10^{-5}$ |
| Train error | 4% | 1.2% | 0.6% | 0.07% | 0% | 0% |
| Validation loss | 0.52 | 0.55 | 0.77 | 0.77 | 1.01 | 1.18 |
| Validation error | 12.4% | 10.4% | 11.1% | 9.1% | 8.92% | 8.9% |

Table 1: Sample values from various epochs in the experiment depicted in Fig. 3.

as we show in Appendix D, if only a single weight layer is being optimized, and furthermore, after a sufficient number of iterations, the activation units stop switching and the training error goes to zero. Second, our results may also be applied directly to the last weight layer if the last hidden layer becomes fixed and linearly separable after a certain number of iterations. This can become true, either approximately, if the input to the last hidden layer is normalized (*e.g.*, using batch norm), or exactly, if the last hidden layer is quantized (Hubara et al., 2016).

## 4.4 MATRIX FACTORIZATION

With multi-layered neural networks in mind, Gunasekar et al. (2017) recently embarked on a study of the implicit bias of under-determined matrix factorization problems, where we minimize the *squared loss* of linear observation of a matrix by gradient descent on its factorization. Since a matrix factorization can be viewed as a two-layer network with linear activations, this is perhaps the simplest deep model one can study in full, and can thus provide insight and direction to studying more complex neural networks. Gunasekar et al. conjectured, and provided theoretical and empirical evidence, that gradient descent on the factorization for an under-determined problem converges to the minimum nuclear norm solution, but only if the initialization is infinitesimally close to zero and the step-sizes are infinitesimally small. With finite step-sizes or finite initialization, Gunasekar et al. could not characterize the bias. It would be interesting to study the same problem with a logistic loss instead of squared loss. Beyond the practical relevance of the logistic loss, taking our approach has the advantage that because of its asymptotic nature, it does not depend on the initialization and step-size. It thus might prove easier to analyze logistic regression on a matrix factorization instead of the least square problem, providing significant insight into the implicit biases of gradient descent on non-convex multi-layered optimization.

## 5 SUMMARY

We characterized the implicit bias induced by gradient descent when minimizing smooth monotone loss functions with an exponential tail. This is the type of loss commonly being minimized in deep learning. We can now rigorously understand:

1. How gradient descent, without early stopping, induces implicit $L_2$ regularization and converges to the maximum $L_2$ margin solution, when minimizing the logistic loss, or exp-loss, or any other monotone decreasing loss with appropriate tail. In particular, the non-tail part does not affect the bias and so the logistic loss and the exp-loss, although very different on non-separable problems, behave the same for separable problems. The bias is also independent of the step-size used (as long as it is small enough to ensure convergence) and (unlike for least square problem) is also independent on the initialization.

2. This convergence is very slow. This explains why it is worthwhile continuing to optimize long after we have zero training error, and even when the loss itself is already extremely small.

3. We should not rely on slow decrease of the training loss, or on no decrease of the validation loss, to decide when to stop. We might improve the validation, and test, errors even when the validation loss increases and even when the decrease in the training loss is tiny.

Perhaps that gradient descent leads to a max $L_2$ margin solution is not a big surprise to those for whom the connection between $L_2$ regularization and gradient descent is natural. Nevertheless, we are not familiar with any prior study or mention of this fact, let alone a rigorous analysis and study of

how this bias is exact and independent of the initial point and the step-size. Furthermore, we also analyze the rate at which this happens, leading to the novel observations discussed above. Perhaps even more importantly, we hope that our analysis can open the door to further analysis of different optimization methods or in different models, including deep networks, where implicit regularization is not well understood even for least square problems, or where we do not have such a natural guess as for gradient descent on linear problems. Analyzing gradient descent on logistic/cross-entropy loss is not only arguably more relevant than the least square loss, but might also be technically easier.

## ACKNOWLEDGMENTS

The authors are grateful to S. Gunasekar, J. Lee, and C. Zeno for helpful comments on the manuscript. The research of DS was supported by the Taub foundation.

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

# Appendix

## A    PROOF OF MAIN RESULTS

In the following proofs, for any solution $\mathbf{w}(t)$, we define

$$\mathbf{r}(t) = \mathbf{w}(t) - \hat{\mathbf{w}} \log t - \tilde{\mathbf{w}},$$

where $\hat{\mathbf{w}}$ and $\tilde{\mathbf{w}}$ follow the conditions of Theorems 3 and 4, that is $\hat{\mathbf{w}}$ is the $L_2$ is the max margin vector, which satisfies eq. 2.4:

$$\hat{\mathbf{w}} = \underset{\mathbf{w} \in \mathbb{R}^d}{\operatorname{argmin}} \|\mathbf{w}\|^2 \text{ s.t. } \forall n : \mathbf{w}^\top \mathbf{x}_n \geq 1,$$

and $\tilde{\mathbf{w}}$ is a vector which satisfies eq. 2.7:

$$\forall n \in \mathcal{S} : \eta \exp\left(-\mathbf{x}_n^\top \tilde{\mathbf{w}}\right) = \alpha_n, \tag{A.1}$$

where we recall that we denoted $\mathbf{X}_{\mathcal{S}} \in \mathbb{R}^{d \times |\mathcal{S}|}$ as the matrix whose columns are the support vectors, a subset $\mathcal{S} \subset \{1, \dots, N\}$ of the columns of $\mathbf{X} = [\mathbf{x}_1, \dots, \mathbf{x}_N] \in \mathbb{R}^{d \times N}$.

In Lemma 8 (Appendix F) we prove that for almost every dataset $\boldsymbol{\alpha}$ is uniquely defined, there are no more then $d$ support vectors and $\alpha_n \neq 0$, $\forall n \in \mathcal{S}$. Therefore, eq. A.1 is well-defined in those cases. If the support vectors do not span the data, then the solution $\tilde{\mathbf{w}}$ to eq. A.1 might not be unique. In this case, we can use any such solution in the proof.

We furthermore denote

$$\theta = \min_{n \notin \mathcal{S}} \mathbf{x}_n^\top \hat{\mathbf{w}} > 1, \tag{A.2}$$

and by $C_i, \epsilon_i, t_i$ ($i \in \mathbb{N}$) various positive constants which are independent of $t$. Lastly, we define $\mathbf{P}_1 \in \mathbb{R}^{d \times d}$ as the orthogonal projection matrix[3] to the subspace spanned by the support vectors (the columns of $\mathbf{X}_{\mathcal{S}}$), and $\mathbf{P}_2 = \mathbf{I} - \mathbf{P}_1$ as the complementary projection (to the left nullspace of $\mathbf{X}_{\mathcal{S}}$).

### A.1    SIMPLE PROOF OF THEOREM 3 FOR A SPECIAL CASE

In this section we first examine the special case that $\ell(u) = e^{-u}$ and take the continuous time limit of gradient descent: $\eta \to 0$, so

$$\dot{\mathbf{w}}(t) = -\nabla \mathcal{L}(\mathbf{w}(t)).$$

The proof in this case is rather short and self-contained (i.e., does not rely on any previous results), and so it helps to clarify the main ideas of the general (more complicated) proof which we will give in the next sections.

Recall we defined

$$\mathbf{r}(t) = \mathbf{w}(t) - \log(t)\hat{\mathbf{w}} - \tilde{\mathbf{w}}. \tag{A.3}$$

Our goal is to show that $\|\mathbf{r}(t)\|$ is bounded, and therefore $\boldsymbol{\rho}(t) = \mathbf{r}(t) + \tilde{\mathbf{w}}$ is bounded. Eq. A.3 implies that

$$\dot{\mathbf{r}}(t) = \dot{\mathbf{w}}(t) - \frac{1}{t}\hat{\mathbf{w}} = -\nabla \mathcal{L}(\mathbf{w}(t)) - \frac{1}{t}\hat{\mathbf{w}} \tag{A.4}$$

and therefore

$$\frac{1}{2}\frac{d}{dt}\|\mathbf{r}(t)\|^2 = \dot{\mathbf{r}}^\top(t)\mathbf{r}(t)$$

$$= \sum_{n=1}^{N} \exp\left(-\mathbf{x}_n^\top \mathbf{w}(t)\right)\mathbf{x}_n^\top \mathbf{r}(t) - \frac{1}{t}\hat{\mathbf{w}}^\top \mathbf{r}(t)$$

$$= \left[\sum_{n \in \mathcal{S}} \exp\left(-\log(t)\hat{\mathbf{w}}^\top \mathbf{x}_n - \tilde{\mathbf{w}}^\top \mathbf{x}_n - \mathbf{x}_n^\top \mathbf{r}(t)\right)\mathbf{x}_n^\top \mathbf{r}(t) - \frac{1}{t}\hat{\mathbf{w}}^\top \mathbf{r}(t)\right]$$

$$+ \left[\sum_{n \notin \mathcal{S}} \exp\left(-\log(t)\hat{\mathbf{w}}^\top \mathbf{x}_n - \tilde{\mathbf{w}}^\top \mathbf{x}_n - \mathbf{x}_n^\top \mathbf{r}(t)\right)\mathbf{x}_n^\top \mathbf{r}(t)\right], \tag{A.5}$$

---

[3]This matrix can be written as $\mathbf{P}_1 = \mathbf{X}_{\mathcal{S}}\mathbf{X}_{\mathcal{S}}^+$, where $\mathbf{M}^+$ is the Moore-Penrose pseudoinverse of $\mathbf{M}$.

where in the last equality we used eq. A.3 and decomposed the sum over support vectors $\mathcal{S}$ and non-support vectors. We examine both bracketed terms.

Recall that $\hat{\mathbf{w}}^\top \mathbf{x}_n = 1$ for $n \in \mathcal{S}$, and that we defined (in eq. A.1) $\tilde{\mathbf{w}}$ so that $\sum_{n \in \mathcal{S}} \exp\left(-\tilde{\mathbf{w}}^\top \mathbf{x}_n\right) \mathbf{x}_n = \hat{\mathbf{w}}$. Thus, the first bracketed term in eq. A.5 can be written as

$$\frac{1}{t} \sum_{n \in \mathcal{S}} \exp\left(-\tilde{\mathbf{w}}^\top \mathbf{x}_n - \mathbf{x}_n^\top \mathbf{r}(t)\right) \mathbf{x}_n^\top \mathbf{r}(t) - \frac{1}{t} \sum_{n \in \mathcal{S}} \exp\left(-\tilde{\mathbf{w}}^\top \mathbf{x}_n\right) \mathbf{x}_n$$

$$= \frac{1}{t} \sum_{n \in \mathcal{S}} \exp\left(-\tilde{\mathbf{w}}^\top \mathbf{x}_n\right) \left(\exp\left(-\mathbf{x}_n^\top \mathbf{r}(t)\right) - 1\right) \mathbf{x}_n^\top \mathbf{r}(t) \leq 0 \tag{A.6}$$

since $z\left(e^{-z} - 1\right) \leq 0$. Furthermore, since $\exp(-z) z \leq 1$ and $\theta = \operatorname{argmin}_{n \notin \mathcal{S}} \mathbf{x}_n^\top \hat{\mathbf{w}}$ (eq. A.2), the second bracketed term in eq. A.5 can be upper bounded by

$$\sum_{n \notin \mathcal{S}} \exp\left(-\log(t) \hat{\mathbf{w}}^\top \mathbf{x}_n - \tilde{\mathbf{w}}^\top \mathbf{x}_n\right) \leq \frac{1}{t^\theta} \sum_{n \notin \mathcal{S}} \exp\left(-\tilde{\mathbf{w}}^\top \mathbf{x}_n\right). \tag{A.7}$$

Substituting eq. A.6 and A.7 into eq. A.5 and integrating, we obtain, that $\exists C, C'$ such that

$$\forall t_1, \forall t > t_1 : \|\mathbf{r}(t)\|^2 - \|\mathbf{r}(t_1)\|^2 \leq C \int_{t_1}^t \frac{dt}{t^\theta} \leq C' < \infty,$$

since $\theta > 1$ (eq. A.2). Thus, we showed that $\mathbf{r}(t)$ is bounded, which completes the proof for the special case. ∎

## A.2 Proof of Theorem 3 (General case)

Next, we give the proof for the general case (discrete time, and exponentially-tailed functions). Though it is based on a similar analysis as in the special case we examined in the previous section, it is somewhat more involved since we have to bound additional terms.

First, we state two auxilary Lemmata, which are proven below in appendix sections A.4 and A.5:

**Lemma 5.** *Let $\mathcal{L}(\mathbf{w})$ be a $\beta$-smooth non-negative objective. If $\eta < 2\beta^{-1}$, then, for any $\mathbf{w}(0)$, with the GD sequence*

$$\mathbf{w}(t+1) = \mathbf{w}(t) - \eta \nabla \mathcal{L}(\mathbf{w}(t)) \tag{A.8}$$

*we have that $\sum_{u=0}^\infty \|\nabla \mathcal{L}(\mathbf{w}(u))\|^2 < \infty$ and therefore $\lim_{t \to \infty} \|\nabla \mathcal{L}(\mathbf{w}(t))\|^2 = 0$.*

**Lemma 6.** *We have*

$$\exists C_1, t_1 : \forall t > t_1 : \left(\mathbf{r}(t+1) - \mathbf{r}(t)\right)^\top \mathbf{r}(t) \leq C_1 t^{-\min(\theta, -1-1.5\mu_+, -1-0.5\mu_-)}. \tag{A.9}$$

*Additionally, $\forall \epsilon_1 > 0$, $\exists C_2, t_2$, such that $\forall t > t_2$, if*

$$\|\mathbf{P}_1 \mathbf{r}(t)\| \geq \epsilon_1, \tag{A.10}$$

*then the following improved bound holds*

$$\left(\mathbf{r}(t+1) - \mathbf{r}(t)\right)^\top \mathbf{r}(t) \leq -C_2 t^{-1} < 0. \tag{A.11}$$

Our goal is to show that $\|\mathbf{r}(t)\|$ is bounded, and therefore $\boldsymbol{\rho}(t) = \mathbf{r}(t) + \tilde{\mathbf{w}}$ is bounded. To show this, we will upper bound the following equation

$$\|\mathbf{r}(t+1)\|^2 = \|\mathbf{r}(t+1) - \mathbf{r}(t)\|^2 + 2\left(\mathbf{r}(t+1) - \mathbf{r}(t)\right)^\top \mathbf{r}(t) + \|\mathbf{r}(t)\|^2 \tag{A.12}$$

First, we note that first term in this equation can be upper-bounded by

$$\|\mathbf{r}(t+1) - \mathbf{r}(t)\|^2$$

$$\overset{(1)}{=} \|\mathbf{w}(t+1) - \hat{\mathbf{w}} \log(t+1) - \tilde{\mathbf{w}} - \mathbf{w}(t) + \hat{\mathbf{w}} \log(t) + \tilde{\mathbf{w}}\|^2$$

$$\overset{(2)}{=} \|-\eta \nabla \mathcal{L}(\mathbf{w}(t)) - \hat{\mathbf{w}} [\log(t+1) - \log(t)]\|^2$$

$$= \eta^2 \|\nabla \mathcal{L}(\mathbf{w}(t))\|^2 + \|\hat{\mathbf{w}}\|^2 \log^2\left(1 + t^{-1}\right) + 2\eta \hat{\mathbf{w}}^\top \nabla \mathcal{L}(\mathbf{w}(t)) \log\left(1 + t^{-1}\right)$$

$$\overset{(3)}{\leq} \eta^2 \|\nabla \mathcal{L}(\mathbf{w}(t))\|^2 + \|\hat{\mathbf{w}}\|^2 t^{-2} \tag{A.13}$$

where in (1) we used eq. 2.3, in (2) we used eq. 2.2, and in (3) we used $\forall x > 0 : x \geq \log(1 + x) > 0$, and also that

$$\hat{\mathbf{w}}^\top \nabla \mathcal{L}\left(\mathbf{w}\left(t\right)\right) = \sum_{n=1}^{N} \ell'\left(\mathbf{w}\left(t\right)^\top \mathbf{x}_n\right) \hat{\mathbf{w}}^\top \mathbf{x}_n \leq 0 , \tag{A.14}$$

since $\hat{\mathbf{w}}^\top \mathbf{x}_n \geq 1$ (from the definition of $\hat{\mathbf{w}}$) and $\ell'(u) \leq 0$.

Also, from Lemma 5 we know that

$$\left\|\nabla \mathcal{L}\left(\mathbf{w}\left(t\right)\right)\right\|^2 = o\left(1\right) \text{ and } \sum_{t=0}^{\infty} \left\|\nabla \mathcal{L}\left(\mathbf{w}\left(t\right)\right)\right\|^2 < \infty . \tag{A.15}$$

Substituting eq. A.15 into eq. A.13, and recalling that a $t^{-\nu}$ power series converges for any $\nu > 1$, we can find $C_0$ such that

$$\left\|\mathbf{r}\left(t+1\right) - \mathbf{r}\left(t\right)\right\|^2 = o\left(1\right) \text{ and } \sum_{t=0}^{\infty} \left\|\mathbf{r}\left(t+1\right) - \mathbf{r}\left(t\right)\right\|^2 = C_0 < \infty . \tag{A.16}$$

Note that this equation also implies that $\forall \epsilon_0$

$$\exists t_0 : \forall t > t_0 : \left|\left\|\mathbf{r}\left(t+1\right)\right\| - \left\|\mathbf{r}\left(t\right)\right\|\right| < \epsilon_0 . \tag{A.17}$$

Next, we would like to bound the second term in eq. A.12. From eq. A.9 in Lemma 6, we can find $t_1, C_1$ such that $\forall t > t_1$:

$$\left(\mathbf{r}\left(t+1\right) - \mathbf{r}\left(t\right)\right)^\top \mathbf{r}\left(t\right) \leq C_1 t^{-\min(\theta, -1-1.5\mu_+, -1-0.5\mu_-)} . \tag{A.18}$$

Thus, by combining eqs. A.18 and A.16 into eq. A.12, we find

$$\begin{aligned}
&\left\|\mathbf{r}\left(t\right)\right\|^2 - \left\|\mathbf{r}\left(t_1\right)\right\|^2 \\
&= \sum_{u=t_1}^{t-1} \left[\left\|\mathbf{r}\left(u+1\right)\right\|^2 - \left\|\mathbf{r}\left(u\right)\right\|^2\right] \\
&\leq C_0 + 2 \sum_{u=t_1}^{t-1} C_1 u^{-\min(\theta, -1-1.5\mu_+, -1-0.5\mu_-)}
\end{aligned}$$

which is a bounded, since $\theta > 1$ (eq. A.2). Therefore, $\left\|\mathbf{r}\left(t\right)\right\|$ is bounded. ∎

## A.3   PROOF OF THEOREM 4

All that remains now is to show that $\left\|\mathbf{r}\left(t\right)\right\| \to 0$ if $\text{rank}\left(\mathbf{X}_\mathcal{S}\right) = \text{rank}\left(\mathbf{X}\right)$, and that $\tilde{\mathbf{w}}$ is unique given $\mathbf{w}\left(0\right)$. To do so, this proof will continue where the proof of Theorem 3 stopped, using notations and equations from that proof.

Since $\mathbf{r}\left(t\right)$ has a bounded norm, its two orthogonal components $\mathbf{r}\left(t\right) = \mathbf{P}_1 \mathbf{r}\left(t\right) + \mathbf{P}_2 \mathbf{r}\left(t\right)$ also have bounded norms (recall that $\mathbf{P}_1, \mathbf{P}_2$ were defined in the beginning of appendix section A). From eq. 2.2, $\nabla \mathcal{L}\left(\mathbf{w}\right)$ is spanned by the columns of $\mathbf{X}$. If $\text{rank}\left(\mathbf{X}_\mathcal{S}\right) = \text{rank}\left(\mathbf{X}\right)$, then it is also spanned by the columns of $\mathbf{X}_\mathcal{S}$, and so $\mathbf{P}_2 \nabla \mathcal{L}\left(\mathbf{w}\right) = 0$. Therefore, $\mathbf{P}_2 \mathbf{r}\left(t\right)$ is not updated during GD, and remains constant. Since $\tilde{\mathbf{w}}$ in eq. 2.3 is also bounded, we can absorb this constant $\mathbf{P}_2 \mathbf{r}\left(t\right)$ into $\tilde{\mathbf{w}}$ without affecting eq. 2.7 (since $\forall n \in \mathcal{S} : \mathbf{x}_n^\top \mathbf{P}_2 \mathbf{r}\left(t\right) = 0$). Thus, without loss of generality, we can assume that $\mathbf{r}\left(t\right) = \mathbf{P}_1 \mathbf{r}\left(t\right)$.

Now, recall eq. A.11 in Lemma 6

$$\exists C_2, t_2 : \forall t > t_2 : \left(\mathbf{r}\left(t+1\right) - \mathbf{r}\left(t\right)\right)^\top \mathbf{r}\left(t\right) \leq -C_2 t^{-1} < 0 .$$

Combining this with eqs. A.12 and A.16, implies that $\exists t_3 > \max\left[t_2, t_0\right]$ such that $\forall t > t_3$ such that $\left\|\mathbf{r}\left(t\right)\right\| > \epsilon_1$, we have that $\left\|\mathbf{r}\left(t+1\right)\right\|^2$ is a decreasing function since then

$$\left\|\mathbf{r}\left(t+1\right)\right\|^2 - \left\|\mathbf{r}\left(t\right)\right\|^2 \leq -C_3 t^{-1} < 0. \tag{A.19}$$

Additionally, this result also implies that we cannot have $\|\mathbf{r}(t)\| > \epsilon_1 \ \forall t > t_3$ , since then we arrive to the contradiction.

$$\|\mathbf{r}(t)\|^2 - \|\mathbf{r}(t_3)\|^2 = \sum_{u=t_3}^{t-1} \left[ \|\mathbf{r}(u+1)\|^2 - \|\mathbf{r}(u)\|^2 \right] \leq - \sum_{u=t_3}^{t-1} C_3 u^{-1} \to -\infty,$$

Therefore, $\exists t_4 > t_3$ such that $\|\mathbf{r}(t_4)\| \leq \epsilon_1$. Recall also that $\|\mathbf{r}(t)\|$ is a decreasing function whenever $\|\mathbf{r}(t)\| \geq \epsilon_1$ (eq. A.19). Also, recall that $t_4 > t_0$, so from eq. A.17, we have that $\forall t > t_4, |\|\mathbf{r}(t+1)\| - \|\mathbf{r}(t)\|| < \epsilon_0$. Combining these three facts we conclude that $\forall t > t_4$ : $\|\mathbf{r}(t)\| \leq \epsilon_1 + \epsilon_0$. Since this reasoning holds $\forall \epsilon_1, \epsilon_0$, this implies that $\|\mathbf{r}(t)\| \to 0$.

Lastly, we note that since $\mathbf{P}_2 \mathbf{r}(t)$ is not updated during GD, we have that $\mathbf{P}_2(\tilde{\mathbf{w}} - \mathbf{w}(0)) = 0$. This sets $\tilde{\mathbf{w}}$ uniquely, together with eq. 2.7. ■

### A.4 PROOF OF LEMMA 5

**Lemma 5.** *Let $\mathcal{L}(\mathbf{w})$ be a $\beta$-smooth non-negative objective. If $\eta < 2\beta^{-1}$, then, for any $\mathbf{w}(0)$, with the GD sequence*

$$\mathbf{w}(t+1) = \mathbf{w}(t) - \eta \nabla \mathcal{L}(\mathbf{w}(t)) \tag{A.8}$$

*we have that $\sum_{u=0}^{\infty} \|\nabla \mathcal{L}(\mathbf{w}(u))\|^2 < \infty$ and therefore $\lim_{t \to \infty} \|\nabla \mathcal{L}(\mathbf{w}(t))\|^2 = 0$.*

This proof is a slightly modified version of the proof of Theorem 2 in (Ganti, 2015). Recall a well-known property of $\beta$-smooth functions:

$$\left| f(\mathbf{x}) - f(\mathbf{y}) - \nabla f(\mathbf{y})^\top (\mathbf{x} - \mathbf{y}) \right| \leq \frac{\beta}{2} \|\mathbf{x} - \mathbf{y}\|^2 . \tag{A.20}$$

From the $\beta$-smoothness of $\mathcal{L}(\mathbf{w})$

$$\mathcal{L}(\mathbf{w}(t+1)) \leq \mathcal{L}(\mathbf{w}(t)) + \nabla \mathcal{L}(\mathbf{w}(t))^\top (\mathbf{w}(t+1) - \mathbf{w}(t)) + \frac{\beta}{2} \|\mathbf{w}(t+1) - \mathbf{w}(t)\|^2$$

$$= \mathcal{L}(\mathbf{w}(t)) - \eta \|\nabla \mathcal{L}(\mathbf{w}(t))\|^2 + \frac{\beta \eta^2}{2} \|\nabla \mathcal{L}(\mathbf{w}(t))\|^2$$

$$= \mathcal{L}(\mathbf{w}(t)) - \eta \left( 1 - \frac{\beta \eta}{2} \right) \|\nabla \mathcal{L}(\mathbf{w}(t))\|^2$$

Thus, we have

$$\frac{\mathcal{L}(\mathbf{w}(t)) - \mathcal{L}(\mathbf{w}(t+1))}{\eta \left( 1 - \frac{\beta \eta}{2} \right)} \geq \|\nabla \mathcal{L}(\mathbf{w}(t))\|^2$$

which implies

$$\sum_{u=0}^{t} \|\nabla \mathcal{L}(\mathbf{w}(u))\|^2 \leq \sum_{u=0}^{t} \frac{\mathcal{L}(\mathbf{w}(u)) - \mathcal{L}(\mathbf{w}(u+1))}{\eta \left( 1 - \frac{\beta \eta}{2} \right)} = \frac{\mathcal{L}(\mathbf{w}(0)) - \mathcal{L}(\mathbf{w}(t+1))}{\eta \left( 1 - \frac{\beta \eta}{2} \right)} .$$

The right hand side is upper bounded by a finite constant, since $L(\mathbf{w}(0)) < \infty$ and $0 \leq \mathcal{L}(\mathbf{w}(t+1))$. This implies

$$\sum_{u=0}^{\infty} \|\nabla \mathcal{L}(\mathbf{w}(u))\|^2 < \infty ,$$

and therefore $\|\nabla \mathcal{L}(\mathbf{w}(t))\|^2 \to 0$.

### A.5 PROOF OF LEMMA 6

Recall that we defined $\mathbf{r}(t) = \mathbf{w}(t) - \hat{\mathbf{w}} \log t - \tilde{\mathbf{w}}$, with $\hat{\mathbf{w}}$ and $\tilde{\mathbf{w}}$ follow the conditions of the Theorems 3 and 4, *i.e.* $\hat{\mathbf{w}}$ is the $L_2$ max margin vector and (eq. 2.4), and eq. 2.7 holds

$$\forall n \in \mathcal{S} : \ \eta \exp \left( -\mathbf{x}_n^\top \tilde{\mathbf{w}} \right) = \alpha_n .$$

**Lemma 6.** *We have*

$$\exists C_1, t_1 : \forall t > t_1 : (\mathbf{r}(t+1) - \mathbf{r}(t))^\top \mathbf{r}(t) \leq C_1 t^{-\min(\theta, -1-1.5\mu_+, -1-0.5\mu_-)}. \tag{A.9}$$

*Additionally,* $\forall \epsilon_1 > 0$, $\exists C_2, t_2$, *such that* $\forall t > t_2$, *if*

$$\|\mathbf{P}_1 \mathbf{r}(t)\| \geq \epsilon_1, \tag{A.10}$$

*then the following improved bound holds*

$$(\mathbf{r}(t+1) - \mathbf{r}(t))^\top \mathbf{r}(t) \leq -C_2 t^{-1} < 0. \tag{A.11}$$

From Lemma 1, $\forall n : \lim_{t\to\infty} \mathbf{w}(t)^\top \mathbf{x}_n = \infty$. In addition, from assumption 3 the negative loss derivative $-\ell'(u)$ has an exponential tail $e^{-u}$ (recall we assume $a = c = 1$ without loss of generality). Combining both facts, we have positive constants $\mu_-, \mu_+, t_-$ and $t_+$ such that $\forall n$

$$\forall t > t_+ : -\ell'\left(\mathbf{w}(t)^\top \mathbf{x}_n\right) \leq \left(1 + \exp\left(-\mu_+ \mathbf{w}(t)^\top \mathbf{x}_n\right)\right) \exp\left(-\mathbf{w}(t)^\top \mathbf{x}_n\right) \tag{A.21}$$

$$\forall t > t_- : -\ell'\left(\mathbf{w}(t)^\top \mathbf{x}_n\right) \geq \left(1 - \exp\left(-\mu_- \mathbf{w}(t)^\top \mathbf{x}_n\right)\right) \exp\left(-\mathbf{w}(t)^\top \mathbf{x}_n\right) \tag{A.22}$$

Next, we examine the expression we wish to bound, recalling that $\mathbf{r}(t) = \mathbf{w}(t) - \hat{\mathbf{w}} \log t - \tilde{\mathbf{w}}$:

$$
\begin{aligned}
&(\mathbf{r}(t+1) - \mathbf{r}(t))^\top \mathbf{r}(t) \\
&= (-\eta \nabla \mathcal{L}(\mathbf{w}(t)) - \hat{\mathbf{w}} [\log(t+1) - \log(t)])^\top \mathbf{r}(t) \\
&= -\eta \sum_{n=1}^N \ell'\left(\mathbf{w}(t)^\top \mathbf{x}_n\right) \mathbf{x}_n^\top \mathbf{r}(t) - \hat{\mathbf{w}}^\top \mathbf{r}(t) \log\left(1 + t^{-1}\right) \\
&= \hat{\mathbf{w}}^\top \mathbf{r}(t) \left[t^{-1} - \log\left(1 + t^{-1}\right)\right] - \eta \sum_{n\notin\mathcal{S}} \ell'\left(\mathbf{w}(t)^\top \mathbf{x}_n\right) \mathbf{x}_n^\top \mathbf{r}(t) \\
&\quad - \eta \sum_{n\in\mathcal{S}} \left[t^{-1} \exp\left(-\tilde{\mathbf{w}}^\top \mathbf{x}_n\right) + \ell'\left(\mathbf{w}(t)^\top \mathbf{x}_n\right)\right] \mathbf{x}_n^\top \mathbf{r}(t)
\end{aligned}
\tag{A.23}
$$

where in last line we used eqs. 2.6 and 2.7 to obtain

$$\hat{\mathbf{w}} = \sum_{n\in\mathcal{S}} \alpha_n \mathbf{x}_n = \eta \sum_{n\in\mathcal{S}} \exp\left(-\tilde{\mathbf{w}}^\top \mathbf{x}_n\right) \mathbf{x}_n.$$

We examine the three terms in eq. A.23. The first term can be upper bounded by

$$
\begin{aligned}
&\hat{\mathbf{w}}^\top \mathbf{r}(t) \left[t^{-1} - \log\left(1 + t^{-1}\right)\right] \\
&\leq \max\left[\hat{\mathbf{w}}^\top \mathbf{r}(t), 0\right] \left[t^{-1} - \log\left(1 + t^{-1}\right)\right] \\
&\overset{(1)}{\leq} \max\left[\hat{\mathbf{w}}^\top \mathbf{P}_1 \mathbf{r}(t), 0\right] t^{-2} \\
&\overset{(2)}{\leq} \begin{cases} \|\hat{\mathbf{w}}\| \epsilon_1 t^{-2} & , \text{ if } \|\mathbf{P}_1 \mathbf{r}(t)\| \leq \epsilon_1 \\ o\left(t^{-1}\right) & , \text{ if } \|\mathbf{P}_1 \mathbf{r}(t)\| > \epsilon_1 \end{cases}
\end{aligned}
\tag{A.24}
$$

where in (1) we used that $\mathbf{P}_2 \hat{\mathbf{w}} = \mathbf{P}_2 \mathbf{X}_\mathcal{S} \alpha = 0$ from eq. 2.6, and in (2) we used that $\hat{\mathbf{w}}^\top \mathbf{r}(t) = o(t)$, since

$$
\begin{aligned}
\hat{\mathbf{w}}^\top \mathbf{r}(t) &= \hat{\mathbf{w}}^\top \left(\mathbf{w}(0) - \eta \sum_{u=0}^t \nabla \mathcal{L}(\mathbf{w}(u)) - \hat{\mathbf{w}} \log(t) - \tilde{\mathbf{w}}\right) \\
&\leq \hat{\mathbf{w}}^\top (\mathbf{w}(0) - \tilde{\mathbf{w}} - \hat{\mathbf{w}} \log(t)) - \eta t \min_{0\leq u\leq t} \hat{\mathbf{w}}^\top \nabla \mathcal{L}(\mathbf{w}(u)) = o(t)
\end{aligned}
$$

where in the last line we used that $\nabla \mathcal{L}(\mathbf{w}(t)) = o(1)$, from Lemma 5.

Next, we upper bound the second term in eq. A.23, $\forall t > t_+$:

$$- \eta \sum_{n \notin \mathcal{S}} \ell' \left( \mathbf{w}(t)^\top \mathbf{x}_n \right) \mathbf{x}_n^\top \mathbf{r}(t)$$

$$\leq - \eta \sum_{n \notin \mathcal{S}: \mathbf{x}_n^\top \mathbf{r}(t) \geq 0} \ell' \left( \mathbf{w}(t)^\top \mathbf{x}_n \right) \mathbf{x}_n^\top \mathbf{r}(t)$$

$$\overset{(1)}{\leq} \eta \sum_{n \notin \mathcal{S}: \mathbf{x}_n^\top \mathbf{r}(t) \geq 0} \left( 1 + \exp \left( -\mu_+ \mathbf{w}(t)^\top \mathbf{x}_n \right) \right) \exp \left( -\mathbf{w}(t)^\top \mathbf{x}_n \right) \mathbf{x}_n^\top \mathbf{r}(t)$$

$$\overset{(2)}{\leq} \eta \sum_{n \notin \mathcal{S}: \mathbf{x}_n^\top \mathbf{r}(t) \geq 0} \left( 1 + t^{-\mu_+ \mathbf{x}_n^\top \hat{\mathbf{w}}} \exp \left( -\mu_+ \tilde{\mathbf{w}}^\top \mathbf{x}_n - \mu_+ \mathbf{x}_n^\top \mathbf{r}(t) \right) \right) t^{-\mathbf{x}_n^\top \hat{\mathbf{w}}} \exp \left( -\tilde{\mathbf{w}}^\top \mathbf{x}_n - \mathbf{x}_n^\top \mathbf{r}(t) \right) \mathbf{x}_n^\top \mathbf{r}(t)$$

$$\overset{(3)}{\leq} \eta \sum_{n \notin \mathcal{S}: \mathbf{x}_n^\top \mathbf{r}(t) \geq 0} \left( 1 + t^{-\mu_+ \mathbf{x}_n^\top \hat{\mathbf{w}}} \exp \left( -\mu_+ \tilde{\mathbf{w}}^\top \mathbf{x}_n \right) \right) t^{-\mathbf{x}_n^\top \hat{\mathbf{w}}} \exp \left( -\tilde{\mathbf{w}}^\top \mathbf{x}_n \right)$$

$$\overset{(4)}{\leq} \eta N \left( 1 + \left[ t^{-\theta} \exp \left( -\min_n \tilde{\mathbf{w}}^\top \mathbf{x}_n \right) \right]^{\mu_+} \right) \exp \left( -\min_n \tilde{\mathbf{w}}^\top \mathbf{x}_n \right) t^{-\theta}$$

$$\overset{(5)}{\leq} 2\eta N \exp \left( -\min_n \tilde{\mathbf{w}}^\top \mathbf{x}_n \right) t^{-\theta}, \forall t > t'_+, \tag{A.25}$$

where in (1) we used eq. A.21, in (2) we used $\mathbf{w}(t) = \hat{\mathbf{w}} \log t + \tilde{\mathbf{w}} + \mathbf{r}(t)$, in (3) we used $xe^{-x} \leq 1$ and $\mathbf{x}_n^\top \mathbf{r}(t) \geq 0$, in (4) we used $\theta > 1$, from eq. A.2 and in (5) we defined $t'_+ = \max \left[ t_+, \exp \left( \min_n \tilde{\mathbf{w}}^\top \mathbf{x}_n \right) \right]$.

Lastly, we will aim to bound the sum in the third term in eq. A.23

$$- \eta \sum_{n \in \mathcal{S}} \left[ t^{-1} \exp \left( -\tilde{\mathbf{w}}^\top \mathbf{x}_n \right) + \ell' \left( \mathbf{w}(t)^\top \mathbf{x}_n \right) \right] \mathbf{x}_n^\top \mathbf{r}(t) . \tag{A.26}$$

We examine each term $k$ in this sum, and divide into two cases, depending on the sign of $\mathbf{x}_k^\top \mathbf{r}(t)$.

First, if $\mathbf{x}_k^\top \mathbf{r}(t) \geq 0$, then term $k$ in eq. A.26 can be upper bounded $\forall t > t_+$, using eq. A.21, by

$$\eta t^{-1} \exp \left( -\tilde{\mathbf{w}}^\top \mathbf{x}_k \right) \left[ \left( 1 + t^{-\mu_+} \exp \left( -\mu_+ \tilde{\mathbf{w}}^\top \mathbf{x}_k \right) \right) \exp \left( -\mathbf{x}_k^\top \mathbf{r}(t) \right) - 1 \right] \mathbf{x}_k^\top \mathbf{r}(t) \tag{A.27}$$

We further divide into cases:

1. If $\left| \mathbf{x}_k^\top \mathbf{r}(t) \right| \leq C_0 t^{-0.5\mu_+}$, then we can upper bound eq. A.27 with

$$\eta \exp \left( -(1 + \mu_+) \min_n \tilde{\mathbf{w}}^\top \mathbf{x}_n \right) C_0 t^{-1-1.5\mu_+} . \tag{A.28}$$

2. If $\left| \mathbf{x}_k^\top \mathbf{r}(t) \right| > C_0 t^{-0.5\mu_+}$, then we can find $t''_+ > t'_+$ to upper bound eq. A.27 $\forall t > t''_+$:

$$\eta t^{-1} e^{-\tilde{\mathbf{w}}^\top \mathbf{x}_k} \left[ \left( 1 + t^{-\mu_+} e^{-\mu_+ \tilde{\mathbf{w}}^\top \mathbf{x}_k} \right) \exp \left( -C_0 t^{-0.5\mu_+} \right) - 1 \right] \mathbf{x}_k^\top \mathbf{r}(t)$$

$$\overset{(1)}{\leq} \eta t^{-1} e^{-\tilde{\mathbf{w}}^\top \mathbf{x}_k} \left[ \left( 1 + t^{-\mu_+} e^{-\mu_+ \tilde{\mathbf{w}}^\top \mathbf{x}_k} \right) \left( 1 - C_0 t^{-0.5\mu_+} + C_0^2 t^{-\mu_+} \right) - 1 \right] \mathbf{x}_k^\top \mathbf{r}(t)$$

$$\leq \eta t^{-1} e^{-\tilde{\mathbf{w}}^\top \mathbf{x}_k} \left[ \left( 1 - C_0 t^{-0.5\mu_+} + C_0^2 t^{-\mu_+} \right) e^{-\mu_+ \min_n \tilde{\mathbf{w}}^\top \mathbf{x}_n} t^{-\mu_+} - C_0 t^{-0.5\mu_+} + C_0^2 t^{-\mu_+} \right] \mathbf{x}_k^\top \mathbf{r}(t)$$

$$\overset{(2)}{\leq} 0, \forall t > t''_+ \tag{A.29}$$

where in (1) we used the fact that $e^{-x} \leq 1 - x + x^2$ for $x \geq 0$ and in (2) we defined $t''_+$ so that the previous expression is negative — this is possible since $t^{-0.5\mu_+}$ decreases slower then $t^{-\mu_+}$.

3. If $\left| \mathbf{x}_k^\top \mathbf{r}(t) \right| \geq \epsilon_2$, then we define $t'''_+ > t''_+$ such that $t'''_+ > \exp \left( \min_n \tilde{\mathbf{w}}^\top \mathbf{x}_n \right) \left[ e^{0.5\epsilon_2} - 1 \right]^{-1/\mu_+}$, and therefore $\forall t > t'''_+$, we have $\left( 1 + t^{-\mu_+} \exp \left( -\mu_+ \tilde{\mathbf{w}}^\top \mathbf{x}_n \right) \right) e^{-\epsilon_2} < e^{-0.5\epsilon_2}$.

   This implies that $\forall t > t'''_+$ we can upper bound eq. A.27 by

$$- \eta \exp \left( -\max_n \tilde{\mathbf{w}}^\top \mathbf{x}_n \right) \left( 1 - e^{-0.5\epsilon_2} \right) \epsilon_2 t^{-1}. \tag{A.30}$$

Second, if $\mathbf{x}_k^\top \mathbf{r}(t) < 0$, we again further divide into cases:

1. If $\left|\mathbf{x}_k^\top \mathbf{r}(t)\right| \leq C_0 t^{-0.5\mu_-}$, then, since $-\ell'\left(\mathbf{w}(t)^\top \mathbf{x}_n\right) > 0$, we can upper bound term $k$ in eq. A.26 with

$$\eta t^{-1} \exp\left(-\tilde{\mathbf{w}}^\top \mathbf{x}_k\right) \left|\mathbf{x}_k^\top \mathbf{r}(t)\right| \leq \eta \exp\left(-\min_n \tilde{\mathbf{w}}^\top \mathbf{x}_n\right) C_0 t^{-1-0.5\mu_-} \qquad \text{(A.31)}$$

2. If $\left|\mathbf{x}_k^\top \mathbf{r}(t)\right| > C_0 t^{-0.5\mu_-}$, then, using eq. A.22 we upper bound term $k$ in eq. A.26 with

$$\eta\left[-t^{-1}e^{-\tilde{\mathbf{w}}^\top \mathbf{x}_k} - \ell'\left(\mathbf{w}(t)^\top \mathbf{x}_k\right)\right] \mathbf{x}_k^\top \mathbf{r}(t)$$
$$\leq \eta\left[-t^{-1}e^{-\tilde{\mathbf{w}}^\top \mathbf{x}_k} + \left(1 - \exp\left(-\mu_- \mathbf{w}(t)^\top \mathbf{x}_k\right)\right) \exp\left(-\mathbf{w}(t)^\top \mathbf{x}_k\right)\right] \mathbf{x}_k^\top \mathbf{r}(t)$$
$$= \eta t^{-1} e^{-\tilde{\mathbf{w}}^\top \mathbf{x}_k}\left[1 - \exp\left(-\mathbf{r}^\top(t)\mathbf{x}_k\right)\left(1 - \left[t^{-1}e^{-\tilde{\mathbf{w}}^\top \mathbf{x}_k} \exp\left(-\mathbf{r}^\top(t)\mathbf{x}_k\right)\right]^{\mu_-}\right)\right] \left|\mathbf{x}_k^\top \mathbf{r}(t)\right|$$
$$\text{(A.32)}$$

Next, we will show that $\exists t'_- > t_-$ such that the last expression is strictly negative $\forall t > t'_-$. Let $M > 1$ be some arbitrary constant. Then, since $\left[t^{-1}e^{-\tilde{\mathbf{w}}^\top \mathbf{x}_k} \exp\left(-\mathbf{r}^\top(t)\mathbf{x}_k\right)\right]^{\mu_-} = \exp\left(-\mu_- \mathbf{w}(t)^\top \mathbf{x}_k\right) \to 0$ from Lemma 1, $\exists t_M > t_-$ such that $\forall t > t_M, t > M e^{-\tilde{\mathbf{w}}^\top \mathbf{x}_k}$, and if $\exp\left(-\mathbf{r}^\top(t)\mathbf{x}_k\right) \geq M > 1$ then

$$\exp\left(-\mathbf{r}^\top(t)\mathbf{x}_k\right)\left(1 - \left[t^{-1}e^{-\tilde{\mathbf{w}}^\top \mathbf{x}_k} \exp\left(-\mathbf{r}^\top(t)\mathbf{x}_k\right)\right]^{\mu_-}\right) \geq M' > 1. \qquad \text{(A.33)}$$

Furthermore, if $\exists t > t_M$ such that $\exp\left(-\mathbf{r}^\top(t)\mathbf{x}_k\right) < M$, then

$$\exp\left(-\mathbf{r}^\top(t)\mathbf{x}_k\right)\left(1 - \left[t^{-1}e^{-\tilde{\mathbf{w}}^\top \mathbf{x}_k} \exp\left(-\mathbf{r}^\top(t)\mathbf{x}_k\right)\right]^{\mu_-}\right)$$
$$> \exp\left(-\mathbf{r}^\top(t)\mathbf{x}_k\right)\left(1 - \left[t^{-1}e^{-\tilde{\mathbf{w}}^\top \mathbf{x}_k} M\right]^{\mu_-}\right). \qquad \text{(A.34)}$$

which is lower bounded by

$$\left(1 + C_0 t^{-0.5\mu_-}\right)\left(1 - t^{-\mu_-}\left[e^{-\tilde{\mathbf{w}}^\top \mathbf{x}_k} M\right]^{\mu_-}\right)$$
$$\geq 1 + C_0 t^{-0.5\mu_-} - t^{-\mu_-}\left[e^{-\tilde{\mathbf{w}}^\top \mathbf{x}_k} M\right]^{\mu_-} - t^{-1.5\mu_-}\left[e^{-\tilde{\mathbf{w}}^\top \mathbf{x}_k} M\right]^{\mu_-} C_0$$

since $\left|\mathbf{x}_k^\top \mathbf{r}(t)\right| > C_0 t^{-0.5\mu_-}$, $\mathbf{x}_k^\top \mathbf{r}(t) < 0$ and $e^x \geq 1 + x$. In this case last line is strictly larger then 1 for sufficiently large $t$. Therefore, after we substitute eqs. A.33 and A.34 into A.32, we find that $\exists t'_- > t_M > t_-$ such that $\forall t > t'_-$, term $k$ in eq. A.26 is strictly negative

$$\eta\left[-t^{-1}e^{-\tilde{\mathbf{w}}^\top \mathbf{x}_k} - \ell'\left(\mathbf{w}(t)^\top \mathbf{x}_k\right)\right] \mathbf{x}_k^\top \mathbf{r}(t) < 0 \qquad \text{(A.35)}$$

3. If $\left|\mathbf{x}_k^\top \mathbf{r}(t)\right| \geq \epsilon_2$, which is a special case of the previous case ($\left|\mathbf{x}_k^\top \mathbf{r}(t)\right| > C_0 t^{-0.5\mu_-}$) then $\forall t > t'_-$, either eq. A.33 or A.34 holds. Furthermore, in this case, $\exists t''_- > t'_-$ and $M'' > 1$ such that $\forall t > t''_-$ eq. A.34 can be lower bounded by

$$\exp(\epsilon_2)\left(1 - \left[t^{-1}e^{-\tilde{\mathbf{w}}^\top \mathbf{x}_k} M\right]^{\mu_-}\right) > M'' > 1.$$

Substituting this, together with eq. A.33, into eq. A.32, we can find $C'_0 > 0$ such we can upper bound term $k$ in eq. A.26 with

$$-C'_0 t^{-1}, \forall t > t''_-. \qquad \text{(A.36)}$$

To conclude, we choose $t_0 = \max\left[t'''_+, t''_-\right]$:

1. If $\|\mathbf{P}_1 \mathbf{r}(t)\| \geq \epsilon_1$ (as in Eq. A.10), we have that

$$\max_{n \in \mathcal{S}} \left|\mathbf{x}_n^\top \mathbf{r}(t)\right|^2 \overset{(1)}{\geq} \frac{1}{|\mathcal{S}|} \sum_{n \in \mathcal{S}} \left|\mathbf{x}_n^\top \mathbf{P}_1 \mathbf{r}(t)\right|^2 = \frac{1}{|\mathcal{S}|} \left\|\mathbf{X}_\mathcal{S}^\top \mathbf{P}_1 \mathbf{r}(t)\right\|^2 \overset{(2)}{\geq} \frac{1}{|\mathcal{S}|} \sigma_{\min}^2\left(\mathbf{X}_\mathcal{S}\right) \epsilon_1^2$$
$$\text{(A.37)}$$

where in (1) we used $\mathbf{P}_1^\top \mathbf{x}_n = \mathbf{x}_n \ \forall n \in \mathcal{S}$, in (2) we denoted by $\sigma_{\min}(\mathbf{X}_\mathcal{S})$, the minimal non-zero singular value of $\mathbf{X}_\mathcal{S}$ and used eq. A.10. Therefore, for some $k$, $\left|\mathbf{x}_k^\top \mathbf{r}\right| \geq \epsilon_2 \triangleq |\mathcal{S}|^{-1} \sigma_{\min}^2(\mathbf{X}_\mathcal{S}) \epsilon_1^2$. In this case, we denote $C_0''$ as the minimum between $C_0'$ (eq. A.36) and $\eta \exp\left(-\max_n \tilde{\mathbf{w}}^\top \mathbf{x}_n\right)\left(1 - e^{-0.5\epsilon_2}\right)\epsilon_2$ (eq. A.30). Then we find that eq. A.26 can be upper bounded by $-C_0'' t^{-1} + o\left(t^{-1}\right)$, $\forall t > t_0$, given eq. A.10. Substituting this result, together with eqs. A.24 and A.25 into eq. A.23, we obtain $\forall t > t_0$

$$\left(\mathbf{r}(t+1) - \mathbf{r}(t)\right)^\top \mathbf{r}(t) \leq -C_0'' t^{-1} + o\left(t^{-1}\right) .$$

This implies that $\exists C_2 < C_0''$ and $\exists t_2 > t_0$ such that eq. A.11 holds. This implies also that eq. A.9 holds for $\|\mathbf{P}_1 \mathbf{r}(t)\| \geq \epsilon_1$.

2. Otherwise, if $\|\mathbf{P}_1 \mathbf{r}(t)\| < \epsilon_1$, we find that $\forall t > t_0$ , each term in eq. A.26 can be upper bounded by either zero (eqs. A.29 and A.35), or terms proportional to $t^{-1-1.5\mu_+}$ (eq. A.28) or $t^{-1-0.5\mu_-}$, (eq. A.31). Combining this together with eqs. A.24, A.25 into eq. A.23 we obtain (for some positive constants $C_3$, $C_4$, $C_5$, and $C_6$)

$$\left(\mathbf{r}(t+1) - \mathbf{r}(t)\right)^\top \mathbf{r}(t) \leq C_3 t^{-1-1.5\mu_+} + C_4 t^{-1-0.5\mu_-} + C_5 t^{-2} + C_6 t^{-\theta} .$$

Therefore, $\exists t_1 > t_0$ and $C_1$ such that eq. A.9 holds. $\square$

## B   CALCULATION OF CONVERGENCE RATES

From Theorem 3, we can write $\mathbf{w}(t) = \hat{\mathbf{w}} \log t + \boldsymbol{\rho}(t)$, where $\boldsymbol{\rho}(t)$ has bounded norm.

Calculation of normalized weight vector (eq. 3.1):

$$\frac{\mathbf{w}(t)}{\|\mathbf{w}(t)\|}$$

$$= \frac{\boldsymbol{\rho}(t) + \hat{\mathbf{w}} \log t}{\sqrt{\boldsymbol{\rho}(t)^\top \boldsymbol{\rho}(t) + \hat{\mathbf{w}}^\top \hat{\mathbf{w}} \log^2 t + 2\boldsymbol{\rho}(t)^\top \hat{\mathbf{w}} \log t}}$$

$$= \frac{\boldsymbol{\rho}(t)/\log t + \hat{\mathbf{w}}}{\|\hat{\mathbf{w}}\| \sqrt{1 + 2\boldsymbol{\rho}(t)^\top \hat{\mathbf{w}}/\left(\|\hat{\mathbf{w}}\|^2 \log t\right) + \|\boldsymbol{\rho}(t)\|^2/\left(\|\hat{\mathbf{w}}\|^2 \log^2 t\right)}}$$

$$= \frac{1}{\|\hat{\mathbf{w}}\|}\left(\boldsymbol{\rho}(t)\frac{1}{\log t} + \hat{\mathbf{w}}\right)\left[1 - \frac{\boldsymbol{\rho}(t)^\top \hat{\mathbf{w}}}{\|\hat{\mathbf{w}}\|^2 \log t} + \left[\frac{3}{4}\left(2\frac{\boldsymbol{\rho}(t)\hat{\mathbf{w}}}{\|\hat{\mathbf{w}}\|^2}\right)^2 - \frac{\|\boldsymbol{\rho}(t)\|^2}{2\|\hat{\mathbf{w}}\|^2}\right]\frac{1}{\log^2 t} + O\left(\frac{1}{\log^3 t}\right)\right] \tag{B.1}$$

$$= \frac{\hat{\mathbf{w}}}{\|\hat{\mathbf{w}}\|} + \left(\frac{\boldsymbol{\rho}(t)}{\|\hat{\mathbf{w}}\|} - \frac{\hat{\mathbf{w}}}{\|\hat{\mathbf{w}}\|}\frac{\boldsymbol{\rho}(t)^\top \hat{\mathbf{w}}}{\|\hat{\mathbf{w}}\|^2}\right)\frac{1}{\log t} + O\left(\frac{1}{\log^2 t}\right)$$

$$= \frac{\hat{\mathbf{w}}}{\|\hat{\mathbf{w}}\|} + \left(\mathbf{I} - \frac{\hat{\mathbf{w}}\hat{\mathbf{w}}^\top}{\|\hat{\mathbf{w}}\|^2}\right)\frac{\boldsymbol{\rho}(t)}{\|\hat{\mathbf{w}}\|}\frac{1}{\log t} + O\left(\frac{1}{\log^2 t}\right)$$

where to obtain eq. B.1 we used $\frac{1}{\sqrt{1+x}} = 1 - \frac{1}{2}x + \frac{3}{4}x^2 + O\left(x^3\right)$, and in the last line we used the fact that $\boldsymbol{\rho}(t)$ has a bounded norm.

We use eq. B.1 to calculate the angle (eq. 3.2):

$$\frac{\mathbf{w}(t)^\top \hat{\mathbf{w}}}{\|\mathbf{w}(t)\| \|\hat{\mathbf{w}}\|}$$

$$= \frac{\hat{\mathbf{w}}^\top}{\|\hat{\mathbf{w}}\|^2}\left(\boldsymbol{\rho}(t)\frac{1}{\log t} + \hat{\mathbf{w}}\right)\left(1 - \frac{1}{\log t}\frac{\boldsymbol{\rho}(t)^\top \hat{\mathbf{w}}}{\|\hat{\mathbf{w}}\|^2} + \left[\frac{3}{4}\left(2\frac{\boldsymbol{\rho}(t)^\top \hat{\mathbf{w}}}{\|\hat{\mathbf{w}}\|^2}\right)^2 - \frac{\|\boldsymbol{\rho}(t)\|^2}{2\|\hat{\mathbf{w}}\|^2}\right]\frac{1}{\log^2 t} + O\left(\frac{1}{\log^3 t}\right)\right)$$

$$= 1 + \frac{2\|\boldsymbol{\rho}(t)\|^2}{\|\hat{\mathbf{w}}\|^2}\left[\left(\frac{\boldsymbol{\rho}(t)^\top \hat{\mathbf{w}}}{\|\hat{\mathbf{w}}\| \|\boldsymbol{\rho}(t)\|}\right)^2 - \frac{1}{4}\right]\frac{1}{\log^2 t} + O\left(\frac{1}{\log^3 t}\right)$$

Calculation of margin (eq. 3.3):

$$\min_n \mathbf{x}_n^\top \hat{\mathbf{w}}(t)$$

$$= \min_n \mathbf{x}_n^\top \left[ \frac{\hat{\mathbf{w}}}{\|\hat{\mathbf{w}}\|} + \left( \frac{\boldsymbol{\rho}(t)}{\|\hat{\mathbf{w}}\|} - \frac{\hat{\mathbf{w}}}{\|\hat{\mathbf{w}}\|} \frac{\boldsymbol{\rho}(t)^\top \hat{\mathbf{w}}}{\|\hat{\mathbf{w}}\|^2} \right) \frac{1}{\log t} + O\left( \frac{1}{\log^2 t} \right) \right]$$

$$= \frac{1}{\|\hat{\mathbf{w}}\|} + \frac{1}{\|\hat{\mathbf{w}}\|} \left( \min_n \mathbf{x}_n^\top \boldsymbol{\rho}(t) - \frac{\boldsymbol{\rho}(t)^\top \hat{\mathbf{w}}}{\|\hat{\mathbf{w}}\|^2} \right) \frac{1}{\log t} + O\left( \frac{1}{\log^2 t} \right) \qquad \text{(B.2)}$$

where in eq. B.2 we used eq. A.2.

Calculation of the training loss (eq. 3.4):

$$\mathcal{L}(\mathbf{w}(t)) \le \sum_{n=1}^N \left( 1 + \exp\left( -\mu_+ \mathbf{w}(t)^\top \mathbf{x}_n \right) \right) \exp\left( -\mathbf{w}(t)^\top \mathbf{x}_n \right)$$

$$= \sum_{n=1}^N \left( 1 + \exp\left( -\mu_+ (\boldsymbol{\rho}(t) + \hat{\mathbf{w}} \log t)^\top \mathbf{x}_n \right) \right) \exp\left( -(\boldsymbol{\rho}(t) + \hat{\mathbf{w}} \log t)^\top \mathbf{x}_n \right)$$

$$= \sum_{n=1}^N \left( 1 + t^{-\mu_+ \hat{\mathbf{w}}^\top \mathbf{x}_n} \exp\left( -\mu_+ \boldsymbol{\rho}(t)^\top \mathbf{x}_n \right) \right) \exp\left( -\boldsymbol{\rho}(t)^\top \mathbf{x}_n \right) t^{-\hat{\mathbf{w}}^\top \mathbf{x}_n}$$

$$= \frac{1}{t} \sum_{n \in \mathcal{S}} e^{-\boldsymbol{\rho}(t)^\top \mathbf{x}_n} + O\left( t^{-\max(\theta, 1+\mu_+)} \right) .$$

Next, we give an example demonstrating the bounds above are strict. Consider optimization with and exponential loss $\ell(u) = e^{-u}$, and a single data point $\mathbf{x} = (1, 0)$. In this case $\hat{\mathbf{w}} = (1, 0)$ and $\|\hat{\mathbf{w}}\| = 1$. We take the limit $\eta \to 0$, and obtain the continuous time version of GD:

$$\dot{w}_1(t) = \exp(-w(t)) \; ; \; \dot{w}_2(t) = 0.$$

We can analytically integrate these equations to obtain

$$w_1(t) = \log(t + \exp(w_1(0))) \; ; \; w_2(t) = w_2(0).$$

Using this example with $w_2(0) > 0$, it is easy to see that the above upper bounds are strict.

Lastly, recall that $\mathcal{V}$ is a set of indices for validation set samples. We calculate of the validation loss for logistic loss, if the error of the $L_2$ max margin vector has some classification errors on the validation, *i.e.,* $\exists k \in \mathcal{V} : \hat{\mathbf{w}}^\top \mathbf{x}_k < 0$:

$$\mathcal{L}_{\text{val}}(\mathbf{w}(t)) = \sum_{n \in \mathcal{V}} \log\left( 1 + \exp\left( -\mathbf{w}(t)^\top \mathbf{x}_n \right) \right)$$

$$\ge \log\left( 1 + \exp\left( -\mathbf{w}(t)^\top \mathbf{x}_k \right) \right)$$

$$= \log\left( 1 + \exp\left( -(\boldsymbol{\rho}(t) + \hat{\mathbf{w}} \log t)^\top \mathbf{x}_k \right) \right)$$

$$= \log\left( \exp\left( -(\boldsymbol{\rho}(t) + \hat{\mathbf{w}} \log t)^\top \mathbf{x}_k \right) \left( 1 + \exp\left( (\boldsymbol{\rho}(t) + \hat{\mathbf{w}} \log t)^\top \mathbf{x}_k \right) \right) \right)$$

$$\ge -(\boldsymbol{\rho}(t) + \hat{\mathbf{w}} \log t)^\top \mathbf{x}_k + \log\left( 1 + \exp\left( (\boldsymbol{\rho}(t) + \hat{\mathbf{w}} \log t)^\top \mathbf{x}_k \right) \right)$$

$$\ge -\log t \, \hat{\mathbf{w}}^\top \mathbf{x}_k + \boldsymbol{\rho}(t)^\top \mathbf{x}_k$$

## C   SOFTMAX OUTPUT WITH CROSS-ENTROPY LOSS

We examine multiclass classification. In the case the labels are the class index $y_n \in \{1, \ldots, K\}$ and we have a weight matrix $\mathbf{W} \in \mathbb{R}^{K \times d}$ with $\mathbf{w}_k$ being the $k$-th row of $\mathbf{W}$.

Furthermore, we define $\mathbf{w} = \text{vec}(\mathbf{W}^\top)$, a basis vector $\mathbf{e}_k \in \mathbb{R}^K$ so that $(\mathbf{e}_k)_i = \delta_{ki}$, and the matrix $\mathbf{A}_k \in \mathbb{R}^{dK \times d}$ so that $\mathbf{A}_k = \mathbf{e}_k \otimes \mathbf{I}_d$, where $\otimes$ is the Kronecker product and $\mathbf{I}_d$ is the $d$-dimension identity matrix. Note that $\mathbf{A}_k^\top \mathbf{w} = \mathbf{w}_k$.

Consider the cross entropy loss with softmax output

$$\mathcal{L}(\mathbf{W}) = -\sum_{n=1}^{N} \log \left( \frac{\exp\left(\mathbf{w}_{y_n}^\top \mathbf{x}_n\right)}{\sum_{k=1}^{K} \exp\left(\mathbf{w}_k^\top \mathbf{x}_n\right)} \right)$$

Using our notation, this loss can be re-written as

$$\mathcal{L}(\mathbf{w}) = -\sum_{n=1}^{N} \log \left( \frac{\exp\left(\mathbf{w}^\top \mathbf{A}_{y_n} \mathbf{x}_n\right)}{\sum_{k=1}^{K} \exp\left(\mathbf{w}^\top \mathbf{A}_k \mathbf{x}_n\right)} \right)$$

$$= \sum_{n=1}^{N} \log \left( \sum_{k=1}^{K} \exp\left(\mathbf{w}^\top \left(\mathbf{A}_k - \mathbf{A}_{y_n}\right) \mathbf{x}_n\right) \right) \tag{C.1}$$

Therefore

$$\nabla \mathcal{L}(\mathbf{w}) = \sum_{n=1}^{N} \frac{\sum_{k=1}^{K} \exp\left(\mathbf{w}^\top \left(\mathbf{A}_k - \mathbf{A}_{y_n}\right) \mathbf{x}_n\right) \left(\mathbf{A}_k - \mathbf{A}_{y_n}\right) \mathbf{x}_n}{\sum_{r=1}^{K} \exp\left(\mathbf{w}^\top \left(\mathbf{A}_r - \mathbf{A}_{y_n}\right) \mathbf{x}_n\right)}$$

$$= \sum_{n=1}^{N} \sum_{k=1}^{K} \frac{1}{\sum_{r=1}^{K} \exp\left(\mathbf{w}^\top \left(\mathbf{A}_r - \mathbf{A}_k\right) \mathbf{x}_n\right)} \left(\mathbf{A}_k - \mathbf{A}_{y_n}\right) \mathbf{x}_n \, .$$

If, again, make the assumption that the data is strictly linearly separable, *i.e.*, in our notation

**Assumption 4.** $\exists \mathbf{w}_*$ *such that* $\mathbf{w}_*^\top \left(\mathbf{A}_k - \mathbf{A}_{y_n}\right) \mathbf{x}_n < 0 \; \forall k \neq y_n$.

then the expression

$$\mathbf{w}_*^\top \nabla \mathcal{L}(\mathbf{w}) = \sum_{n=1}^{N} \sum_{k=1}^{K} \frac{\mathbf{w}_*^\top \left(\mathbf{A}_k - \mathbf{A}_{y_n}\right) \mathbf{x}_n}{\sum_{r=1}^{K} \exp\left(\mathbf{w}^\top \left(\mathbf{A}_r - \mathbf{A}_k\right) \mathbf{x}_n\right)} \, .$$

is strictly negative for any finite $\mathbf{w}$. However, from Lemma 5, in gradient descent with learning rate $\eta > 2\beta^{-1}$, we have that $\nabla L(\mathbf{w}(t)) \to \mathbf{0}$. This implies that: $\|\mathbf{w}(t)\| \to \infty$, and $\forall k \neq y_n, \exists r : \mathbf{w}(t)^\top \left(\mathbf{A}_r - \mathbf{A}_k\right) \mathbf{x}_n \to \infty$, which implies $\forall k \neq y_n, \max_k \mathbf{w}(t)^\top \left(\mathbf{A}_k - \mathbf{A}_{y_n}\right) \mathbf{x}_n \to -\infty$. Examining the loss (eq. C.1) we find that $\mathcal{L}(\mathbf{w}(t)) \to \mathbf{0}$ in this case. Thus, we arrive to an equivalent Lemma to Lemma 1, for this case:

**Lemma 7.** *Let* $\mathbf{w}(t)$ *be the iterates of gradient descent (eq. 2.2) with* $\eta < 2\beta^{-1}$, *for cross-entropy loss operating on a softmax output, under the assumption of strict linear separability (Assumption 4), then: (1)* $\lim_{t \to \infty} \mathcal{L}(\mathbf{w}(t)) = 0$, *(2)* $\lim_{t \to \infty} \|\mathbf{w}(t)\| = \infty$, *and (3)* $\forall n, k \neq y_n : \lim_{t \to \infty} \mathbf{w}(t)^\top \left(\mathbf{A}_{y_n} - \mathbf{A}_k\right) \mathbf{x}_n = \infty$.

Therefore, since

$$\mathcal{L}(\mathbf{w}(t)) = \sum_{n=1}^{N} \log \left( \sum_{k=1}^{K} \exp\left(\mathbf{w}(t)^\top \left(\mathbf{A}_k - \mathbf{A}_{y_n}\right) \mathbf{x}_n\right) \right)$$

$$\approx \sum_{n=1}^{N} \log \left( 1 + \max_{k \neq y_n} \exp\left(\mathbf{w}(t)^\top \left(\mathbf{A}_k - \mathbf{A}_{y_n}\right) \mathbf{x}_n\right) \right) , \tag{C.2}$$

where in the last line we assumed for simplicity that $\mathbf{w}(t)^\top \left(\mathbf{A}_k - \mathbf{A}_{y_n}\right) \mathbf{x}_n$ has a unique minimum in $k$, since then the other exponential terms inside the log become negligible. If

$$\underset{k \neq y_n}{\arg\max} \exp\left(\mathbf{w}(t)^\top \left(\mathbf{A}_k - \mathbf{A}_{y_n}\right) \mathbf{x}_n\right)$$

has a limit $k_n$, then we define $\tilde{\mathbf{x}}_n = (\mathbf{A}_{y_n} - \mathbf{A}_{k_n}) \mathbf{x}_n$, so eq. C.2 is transformed to the standard logistic regression loss

$$\sum_{n=1}^{N} \log \left( 1 + \exp \left( -\mathbf{w}(t)^\top \tilde{\mathbf{x}}_n \right) \right),$$

to which our Theorems directly apply.

Therefore, $\mathbf{w}(t) / \|\mathbf{w}(t)\| \to \hat{\mathbf{w}}$ where

$$\hat{\mathbf{w}} = \underset{\mathbf{w}}{\operatorname{argmin}} \|\mathbf{w}\|^2 \ \text{s.t.} \ \forall n : \mathbf{w}^\top \tilde{\mathbf{x}}_n \geq 1$$

Recalling that $\mathbf{A}_k^\top \mathbf{w} = \mathbf{w}_k$, we can re-write this as

$$\arg \min_{\mathbf{w}_1, \ldots, \mathbf{w}_K} \sum_{k=1}^{K} \|\mathbf{w}_k\|^2 \ \text{s.t.} \ \forall n, \forall k \neq y_n : \mathbf{w}_{y_n}^\top \mathbf{x}_n \geq \mathbf{w}_k^\top \mathbf{x}_n + 1$$

## D  DEEP NETWORKS, IF ONLY A SINGLE LAYER IS OPTIMIZED

We examine a deep neural network (DNN) with $m = 1, \ldots, L$ layers, piecewise linear activation functions $\mathbf{f}_l$ and loss function $\ell$ following assumption 3, parameterized by weights matrices $\mathbf{W}_l$. Since $\mathbf{f}_l$ are piecewise linear, we can write for almost every $\mathbf{u}$: $\mathbf{f}_l(\mathbf{u}) = \nabla \mathbf{f}_l(\mathbf{u}) \odot \mathbf{u}$ (an element-wise product). Given an input sample $\mathbf{x}_n$, for each layer $l$ the input $\mathbf{u}_{n,l}$ and output $\mathbf{v}_{n,l}$ are calculated sequentially in a "forward propagation"

$$\mathbf{u}_{n,l} = \mathbf{W}_l \mathbf{v}_{n,l-1} \ ; \ \mathbf{v}_{n,l} = \mathbf{f}_l(\mathbf{u}_{n,l}) \tag{D.1}$$

initialized by $\mathbf{v}_{n,0} = \mathbf{x}_n$. Then, given the DNN output $u_{n,L}$ and target $y_n \in \{-1, 1\}$ the loss $\ell(y_n u_{n,L})$ can be calculated.

During training, the gradients of the loss are calculated using the chain rule in a "back-propagation"

$$\boldsymbol{\delta}_{n,l-1} = [\nabla \mathbf{f}_l(\mathbf{u}_{n,l})] \mathbf{W}_l^\top \boldsymbol{\delta}_{n,l} \tag{D.2}$$

initialized by $\delta_{n,L} = 1$. Finally, the weights are updated with GD. The basic update (without weight sharing) is

$$\mathbf{W}_l(t+1) - \mathbf{W}_l(t) = -\eta \sum_{n=1}^{N} \frac{\partial}{\partial \mathbf{W}_l} \ell(y_n u_{n,L}) \tag{D.3}$$

$$= -\eta \sum_{n=1}^{N} y_n \ell'(y_n u_{n,L}) \boldsymbol{\delta}_{n,l} \mathbf{v}_{n,l-1}^\top \tag{D.4}$$

$$= -\eta \sum_{n=1}^{N} y_n \ell'\left( y_n \boldsymbol{\delta}_{n,l}^\top \mathbf{W}_l \mathbf{v}_{n,l-1} \right) \boldsymbol{\delta}_{n,l} \mathbf{v}_{n,l-1}^\top , \tag{D.5}$$

where in the last line we used

$$\forall l : \ u_{n,L} = \mathbf{W}_L \left( \prod_{m=1}^{L-1} \nabla \mathbf{f}_l(\mathbf{u}_{n,m}) \mathbf{W}_m \right) \mathbf{x}_n = \boldsymbol{\delta}_{n,l}^\top \mathbf{W}_l \mathbf{v}_{n,l-1} .$$

Denoting $\tilde{\mathbf{x}}_{n,l} = y_n \boldsymbol{\delta}_{n,l} \otimes \mathbf{v}_{n,l-1}$ and $\mathbf{w}_l = \operatorname{vec}\left( \mathbf{W}_l^\top \right)$ we obtain

$$\mathbf{w}_l(t+1) - \mathbf{w}_l(t) = -\eta \sum_{n=1}^{N} \ell'\left( \mathbf{w}_l^\top \tilde{\mathbf{x}}_{n,l} \right) \tilde{\mathbf{x}}_{n,l} .$$

We got the same update as in eq. 2.2. Thus, if $\tilde{\mathbf{x}}_{n,l}$ does not change between iterations and becomes linearly separable so the training error can go to zero, we can apply Theorem 3. This can happen if we only optimize $\mathbf{W}_l$, and the activation units stop crossing their thresholds, after a sufficient number of iterations.

# E    AN EXPERIMENT WITH STOCHASTIC GRADIENT DESCENT

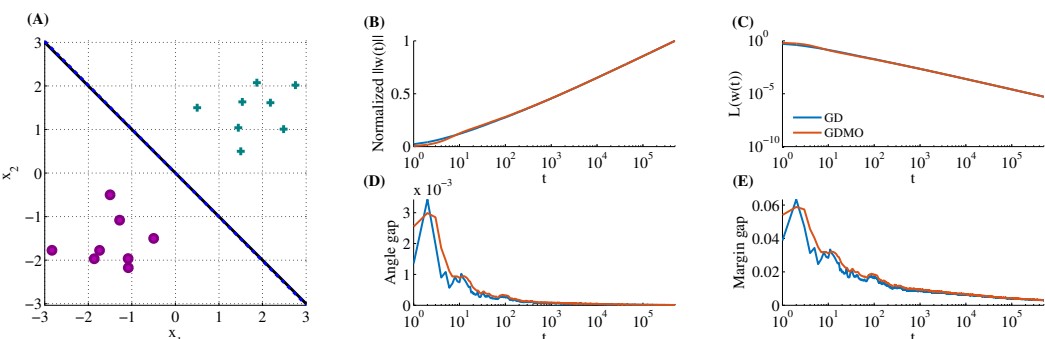

Figure 4: Same as Fig. 1, except stochastic gradient decent is used (with mini-batch of size 4), instead of GD.

# F    GENERIC SOLUTIONS OF THE KKT CONDITIONS IN EQ. 2.6

**Lemma 8.** *For almost all datasets there is a unique $\boldsymbol{\alpha}$ which satisfies the KKT conditions (eq. 2.6):*

$$\hat{\mathbf{w}} = \sum_{n=1}^{N} \alpha_n \mathbf{x}_n \qquad \forall n \ \left( \alpha_n \geq 0 \ \text{and} \ \hat{\mathbf{w}}^\top \mathbf{x}_n = 1 \right) \ \ OR \ \ \left( \alpha_n = 0 \ \text{and} \ \hat{\mathbf{w}}^\top \mathbf{x}_n > 1 \right)$$

*Furthermore, in this solution $\alpha_n \neq 0$ if $\hat{\mathbf{w}}^\top \mathbf{x}_n = 1$, i.e., $\mathbf{x}_n$ is a support vector ($n \in \mathcal{S}$), and there are at most $d$ such support vectors.*

*Proof.* For almost every set $\mathbf{X}$, no more than $d$ points $\mathbf{x}_n$ can be on the same hyperplane. Therefore, since all support vectors must lie on the same hyperplane, there can be at most $d$ support vectors, for almost every $\mathbf{X}$.

Given the set of support vectors, $\mathcal{S}$, the KKT conditions of eq. 2.6 entail that $\alpha_n = 0$ if $n \notin \mathcal{S}$ and

$$1 = \mathbf{X}_{\mathcal{S}}^\top \hat{\mathbf{w}} = \mathbf{X}_{\mathcal{S}}^\top \mathbf{X}_{\mathcal{S}} \boldsymbol{\alpha}_{\mathcal{S}} \,, \tag{F.1}$$

where we denoted $\boldsymbol{\alpha}_{\mathcal{S}}$ as $\boldsymbol{\alpha}$ restricted to the support vector components. For almost every set $\mathbf{X}$, since $d \geq |\mathcal{S}|$, $\mathbf{X}_{\mathcal{S}}^\top \mathbf{X}_{\mathcal{S}} \in \mathbb{R}^{|\mathcal{S}| \times |\mathcal{S}|}$ is invertible. Therefore, $\boldsymbol{\alpha}_{\mathcal{S}}$ has the unique solution

$$\left( \mathbf{X}_{\mathcal{S}}^\top \mathbf{X}_{\mathcal{S}} \right)^{-1} \mathbf{1} = \boldsymbol{\alpha}_{\mathcal{S}} \,. \tag{F.2}$$

This implies that $\forall n \in \mathcal{S}$, $\alpha_n$ is equal to a rational function in the components of $\mathbf{X}_S$, *i.e.*, $\alpha_n = p_n (\mathbf{X}_{\mathcal{S}}) / q_n (\mathbf{X}_{\mathcal{S}})$, where $p_n$ and $q_n$ are polynomials in the components of $\mathbf{X}_S$. Therefore, if $\alpha_n = 0$, then $p_n (\mathbf{X}_{\mathcal{S}}) = 0$, so the components of $\mathbf{X}_{\mathcal{S}}$ must be at a root of the polynomial $p_n$. The roots of the polynomial $p_n$ have measure zero, unless $\forall \mathbf{X}_{\mathcal{S}} : \ p_n (\mathbf{X}_{\mathcal{S}}) = 0$. However, $p_n$ cannot be identically equal to zero, since, for example, if $\mathbf{X}_{\mathcal{S}}^\top = \left[ \mathbf{I}_{|\mathcal{S}| \times |\mathcal{S}|}, \mathbf{0}_{|\mathcal{S}| \times (d-|\mathcal{S}|)} \right]$, then $\mathbf{X}_{\mathcal{S}}^\top \mathbf{X}_{\mathcal{S}} = \mathbf{I}_{|\mathcal{S}| \times |\mathcal{S}|}$, and so in this case $\forall n \in \mathcal{S}$, $\alpha_n = 1 \neq 0$, from eq. F.2.

Therefore, for a given $\mathcal{S}$, the event that "eq. F.1 has a solution with a zero component" has a zero measure. Moreover, the union of these events, for all possible $\mathcal{S}$, also has zero measure, as a finite union of zero measures sets (there are only finitely many possible sets $\mathcal{S} \subset \{1, \ldots, N\}$ ). This implies that, for almost all datasets $\mathbf{X}$, $\alpha_n = 0$ only if $n \notin \mathcal{S}$. Furthermore, for almost all datasets the solution $\boldsymbol{\alpha}$ is unique: for each dataset, $\mathcal{S}$ is uniquely deteremined, and given $\mathcal{S}$ , the solution eq. F.1 is uniquely given by eq. F.2. $\square$

