# OpenReview forum: "The Implicit Bias of Gradient Descent on Separable Data"
_ICLR.cc/2018/Conference — Accept (Poster)_

### Official Review · AnonReviewer3 · 2017-11-24
**An interesting paper, but issues with correctness and presentation**

**Rating:** 5
**Confidence:** 5

**Review:**

The paper offers a formal proof that gradient descent on the logistic
loss converges very slowly to the hard SVM solution in the case where
the data are linearly separable. This result should be viewed in the
context of recent attempts at trying to understand the generalization
ability of neural networks, which have turned to trying to understand
the implicit regularization bias that comes from the choice of
optimizer. Since we do not even understand the regularization bias of
optimizers for the simpler case of linear models, I consider the paper's
topic very interesting and timely.

The overall discussion of the paper is well written, but on a more
detailed level the paper gives an unpolished impression, and has many
technical issues. Although I suspect that most (or even all) of these
issues can be resolved, they interfere with checking the correctness of
the results. Unfortunately, in its current state I therefore do not
consider the paper ready for publication.


Technical Issues:

The statement of Lemma 5 has a trivial part and for the other part the
proof is incorrect: Let x_u = ||nabla L(w(u))||^2.
  - Then the statement sum_{u=0}^t x_u < infinity is trivial, because
    it follows directly from ||nabla L(w(u))||^2 < infinity for all u. I
    would expect the intended statement to be sum_{u=0}^infinity x_u <
    infinity, which actually follows from the proof of the lemma.
  - The proof of the claim that t*x_t -> 0 is incorrect: sum_{u=0}^t x_u
    < infinity does not in itself imply that t*x_t -> 0, as claimed. For
    instance, we might have x_t = 1/i^2 when t=2^i for i = 1,2,... and
    x_t = 0 for all other t.

Definition of tilde{w} in Theorem 4:
  - Why would tilde{w} be unique? In particular, if the support vectors
    do not span the space, because all data lie in the same
    lower-dimensional hyperplane, then this is not the case.
  - The KKT conditions do not rule out the case that \hat{w}^top x_n =
    1, but alpha_n = 0 (i.e. a support vector that touches the margin,
    but does not exert force against it). Such n are then included in
    cal{S}, but lead to problems in (2.7), because they would require
    tilde{w}^top x_n = infinity, which is not possible.

In the proof of Lemma 6, case 2. at the bottom of p.14:
  - After the first inequality, C_0^2 t^{-1.5 epsilon_+} should be
    C_0^2 t^{-epsilon_+}
  - After the second inequality the part between brackets is missing an
    additional term C_0^2 t^{-\epsilon_+}.
  - In addition, the label (1) should be on the previous inequality and
    it should be mentioned that e^{-x} <= 1-x+x^2 is applied for x >= 0
    (otherwise it might be false).
In the proof of Lemma 6, case 2 in the middle of p.15:
  - In the line of inequality (1) there is a t^{-epsilon_-} missing. In
    the next line there is a factor t^{-epsilon_-} too much.
  - In addition, the inequality e^x >= 1 + x holds for all x, so no need
    to mention that x > 0.

In Lemma 1:
  - claim (3) should be lim_{t \to \infty} w(t)^\top x_n = infinity
  - In the proof: w(t)^top x_n > 0 only holds for large enough t.

Remarks:

p.4 The claim that "we can expect the population (or test)
misclassification error of w(t) to improve" because "the margin of w(t)
keeps improving" is worded a little too strongly, because it presumes
that the maximum margin solution will always have the best
generalization error.

In the proof sketch (p.3):
  - Why does the fact that the limit is dominated by gradients that are
    a linear combination of support vectors imply that w_infinity will
    also be a non-negative linear combination of support vectors?
  - "converges to some limit". Mention that you call this limit
    w_infinity


Minor Issues:

In (2.4): add "for all n".

p.10, footnote: Shouldn't "P_1 = X_s X_s^+" be something like "P_1 =
(X_s^top X_s)^+"?

A.9: ell should be ell'

The paper needs a round of copy editing. For instance:
  - top of p.4: "where tilde{w} A is the unique"
  - p.10: "the solution tilde{w} to TO eq. A.2"
  - p.10: "might BOT be unique"
  - p.10: "penrose-moorse pseudo inverse" -> "Moore-Penrose
    pseudoinverse"

In the bibliography, Kingma and Ba is cited twice, with different years.

---

> ### Author Response · Authors · 2017-12-20
> **Comments addressed in revision**
>
> We thank the reviewer for acknowledging the significance of our results, and for investing significant efforts in improving the quality of this manuscript. We uploaded a revised version in which all the reviewer comments were addressed, and the appendix was further polished. Notably,
>
> [Lemma 5 in appdendix]
>
> - Indeed, the upper limit of the sum over x_u should be 'infinity' instead of 't'.
>
> - It should be 'x_t -> 0', not 't*x_t -> 0'.
>
> [Definition of tilde{w} Theorem 4]
>
> - tilde{w} is indeed unique, given the initial conditions. We clarified this in Theorem 4 and its proof.
>
> - alpha_n=0 for the support vectors is only true for a measure zero of all datasets (we added a proof of this in appendix F). Thus, we clarified in the revision that our results hold for almost every dataset (and so, they are true with probability 1 for any data drawn from a continuous-valued distribution).
>
> [Why does the fact that the limit is dominated by gradients that are a linear combination of support vectors imply that w_infinity will also be a non-negative linear combination of support vectors?]
>
> We clarified in the revision: “...The negative gradient would then asymptotically become a non-negative linear combination of support vectors. The limit w_{\infinity} will then be dominated by these gradients, since any initial conditions become negligible as ||w(t)||->infinity (from Lemma 1)”.

---

### Official Review · AnonReviewer1 · 2017-11-25
**Very interesting characterisation of limiting behaviour of the log-loss minimisaton**

**Rating:** 7
**Confidence:** 4

**Review:**

Paper focuses on characterising behaviour of the log loss minimisation on the linearly separable data. As we know, optimisation like this does not converge in a strict mathematical sense, as the norm of the model will grow to infinity. However, one can still hope for a convergence of normalised solution (or equivalently - convergence in term of separator angle, rather than parametrisation). This paper shows that indeed, log-loss (and some other similar losses), minimised with gradient descent, leads to convergence (in the above sense) to the max-margin solution. On one hand it is an interesting property of model we train in practice, and on the other - provides nice link between two separate learning theories.

Pros:
- easy to follow line of argument
- very interesting result of mapping "solution" of unregularised logistic regression (under gradient descent optimisation) onto hard max margin one

Cons:
- it is not clear in the abstract, and beginning of the paper what "convergence" means, as in the strict sense logistic regression optimisation never converges on separable data. It would be beneficial for the clarity if authors define what they mean by convergence (normalised weight vector, angle, whichever path seems most natural) as early in the paper as possible.

---

> ### Author Response · Authors · 2017-12-20
> **Comment addressed in revision**
>
> We thank the reviewer for the positive review and for the helpful comment. We uploaded a revised version in which clarified in the abstract that the weights converge “in direction” to the L2 max margin solution.

---

### Official Review · AnonReviewer2 · 2017-11-27
**This paper analyzes the implicit regularization introduced by gradient descent for optimizing the smooth monotone exponential tailed loss function with separable data. The proposed result is very interesting since it illustrates that using gradient descent to minimize such loss function can lead to the L_2 maximum margin separator.**

**Rating:** 8
**Confidence:** 4

**Review:**

(a) Significance
The main contribution of this paper is to characterize the implicit bias introduced by gradient descent on separable data. The authors show the exact form of this bias (L_2 maximum margin separator), which is independent of the initialization and step size. The corresponding slow convergence rate explains the phenomenon that the predictor can continue to improve even when the training loss is already small. The result of this paper can inspire the study of the implicit bias introduced by gradient descent variants or other optimization methods, such as coordinate descent. In addition, the proposed analytic framework seems promising since it may be extended to analyze other models, like neural networks.

(b) Originality
This is the first work to give the detailed characterizations of the implicit bias of gradient descent on separable data. The proposed assumptions are reasonable, but it seems to limit to the loss function with exponential tail. I’m curious whether the result in this paper can be applied to other loss functions, such as hinge loss.

(c) Clarity & Quality
The presentation of this paper is OK. However, there are some places can be improved in this paper. For example, in Lemma 1, results (3) and (4) can be combined together. It is better for the authors to use another section to illustrate experimental settings instead of writing them in the caption of Figure 3.1.

Minor comments:
1. In Lemma 1 (4), w^T(t)->w(t)^T
2. In the proof of Lemma 1, it’s better to use vector 0 for the gradient L(w)
3. In Theorem 4, the authors should specify eta
4. In appendix A, page 11, beta is double used
5. In appendix D, equation (D.5) has an extra period

---

> ### Author Response · Authors · 2017-12-20
> **Comments addressed in revision**
>
> We thank the reviewer for the positive review and for the helpful comments. We uploaded a revised version in which all the reviewer comments were addressed.
>
> [“I’m curious whether the result in this paper can be applied to other loss functions, such as hinge loss.”]
>
> We believe our results could be extended to many other types of loss functions (in fact, we are currently working on such extensions). However, for the hinge loss (without regularization), gradient descent on separable data can converge to a finite solution which is not to the max margin vector. For example, if there is a single data point x=(1,0), and we start with a weight vector w=(2,2), the hinge loss and its gradient are both equal to zero. Therefore, no weight updates are performed, and we do not converge to the direction of the L2 max margin classifier: w=(1,0).
>
> [“It is better for the authors to use another section to illustrate experimental settings instead of writing them in the caption of Figure 3.1. “]
>
> We felt it is easier to read if all details are summarized in the figure, and wanted to save space to fit the main paper into 8 pages. However, we can change this if required.

---

### Decision · Program_Chairs · 2018-01-29
**ICLR 2018 Conference Acceptance Decision**

**Decision:**

Accept (Poster)

**Comment:**

The paper is tackling an important open problem.

AnonReviewer3 identified some technical issues that led them to rate the manuscript 5 (i.e., just below the acceptance threshold). Many of these issues are resolved by the reviewer in their review, and the author response makes it clear that these fixes are indeed correct.  However, other issues that the reviewer raises are not provided with solutions.  The authors address these points, but in one case at least (regarding w_infinity), I find the new text somewhat hand-waivy. Regardless, I'm inclined to accept the paper because the issues seem to be straightforward. Ultimately, the authors are responsible for the correctness of the results.